# The inflammatory path toward type 1 diabetes begins during pregnancy

Angelica P. Ahrens [1], Raquel Dias[1], Tuulia Hyötyläinen [2], Matej Orešič [3,4,5], Eric W. Triplett [1] ✉ & Johnny Ludvigsson [6] ✉

Type 1 diabetes (T1D) is increasing globally, yet the earliest biological determinants remain poorly defined, particularly in general population studies. We studied the Swedish population-based ABIS birth cohort (n = 16,683) to identify early-life risk factors. Olink proteomic analysis (n = 286 controls, n = 146 cases) of inflammatory signals at birth shows differential abundance years before diagnosis (mean age 12.6 years), with proteins enriched for neutrophil migration, cytotoxicity, extracellular matrix remodeling, and immune regulation. Several markers remain significant in spite of prenatal and perinatal factors including family history of diabetes, and are associated with differences in compounds like stearic acid, lysine, glutamine, and persistent, environmental toxicants perfluorodecylethanoic acid and perfluorooctane sulfonate (PFOS). Using machine learning, we identify a protein subset that predicts T1D with high accuracy (AUC = 0.89 ± 0.02), independently of HLA genetic risk. These findings suggest that innate and tissue-remodeling pathways are perturbed at birth, possibly reflecting early β-cell vulnerability. Identifying these disruptions at birth with a non-invasive method opens a window for prevention, protecting β-cells before the inflammatory attack on islets begins.

Type 1 diabetes (T1D) is a chronic autoimmune disease in which the β-cells of the pancreas are destroyed, ultimately leading to insulin deficiency. An estimated 8.4 million people worldwide live with T1D, with prevalence projected to increase to 13.5–17.4 million in the next two decades[1]. T1D can lead to acute and late complications. Environmental factors are associated with increased incidence[2,3], including childhood enteroviral infections[4–7], maternal respiratory or enteric infections during pregnancy[4], serious life events[5,6], and psychological stress[7,8]. The autoimmune process and risk of progression to clinical diabetes are strongly linked to genetics, including human leukocyte antigen (HLA) DR-DQ genotype. HLA genes, especially class II, contribute to 40–50% of genetic risk[9,10], although less consequential if multiple autoantibodies have already developed[11].

T1D etiology may be driven by autoimmunity and/or by increased insulin demand, triggering glucotoxicity and the other events that ultimately lead to β-cell destruction[1,12,13] is not known. β-cells, compared to α-cells, are more sensitive to inflammatory conditions, where exposure to the immune system and cytokines causes significant gene and protein expression changes. These changes include activation of signal transducer and activator of transcription 1, interferon regulatory factor 1, and nuclear factor kappa-light-chain-enhancer of activated B cells (NF-κB) downstream pathways and adaptive, compensatory mechanisms that are activated to help deal with the inflammatory pressure[12]. Environmental risk factors for T1D continue to be explored, but findings often vary across observational cohorts, complicating the identification of specific triggers[14]. Issues such as selection bias in

[1]Department of Microbiology and Cell Science, Institute of Food and Agricultural Sciences, University of Florida, Gainesville, FL, USA. [2]School of Science and Technology, Örebro University, Örebro, Sweden. [3]School of Medical Sciences, Faculty of Medicine and Health, Örebro University, Örebro, Sweden. [4]Turku Bioscience Centre, University of Turku and Åbo Akademi University, Turku, Finland. [5]Department of Life Technologies, University of Turku, Turku, Finland. [6]Crown Princess Victoria Children's Hospital and Division of Pediatrics, Department of Biomedical and Clinical Sciences, Linköping University, Linköping, SE, Sweden. ✉e-mail: ewt@ufl.edu; johnny.ludvigsson@liu.se

case-control studies and the limitations of autoantibody-based prediction further challenge interpretation[15]. Ultimately, current evidence suggests that immune tolerance is disrupted by varied and diverse environmental triggers and exposures, leading to β-cell stress[3].

HLA genetics[16] and autoantibodies are useful in prediction[17]. Non-invasive biomarkers, irrespective of genetic risk[18] would be a major advance. Proteomic studies in high-genetic-risk children[19,20] have revealed changes near seroconversion and before disease onset (after autoimmunity)[21], with notable temporal shifts just before autoantibody detection. While these findings lay important groundwork and suggest contributors to beta-cell destruction, they are limited either by small sample size or a focus on later disease stages, leaving the earliest markers, such as those present at birth, unexplored[22].

Hence, we investigate inflammatory birth proteomic markers in a large, general population cohort. Proteomic signatures are identified, suggesting that pathogenesis begins as early as pregnancy. Proteomics may offer significant prediction potential in the general population, without reliance on genetic screening, islet antibodies assays, or any invasive procedure.

## Results

The All Babies in Southeast Sweden (ABIS)[23] general population, birth cohort of 16,683 infants is used in this study and represents 78.6% of all children born in Östergötland, Småland, Blekinge, and Öland counties from Oct 1st 1997–Oct 1st 1999 (Fig. 1a). Physician diagnoses (ICD-10 code, E10) are obtained from the National Swedish Patient Register, validated by the National Swedish Drug Prescription Register, inclusive of all diagnosed cases up until Dec 2023. Cases included 167 ABIS children are diagnosed T1D up until this time, with a cumulative incidence of 1% and age at diagnosis ranging from 2 to 24.6 years (mean ± SD: 12.6 ± 6.1 years, median: 12.6 years, 95% CI [11.7, 13.5]; Fig. 1f). Here, controls are defined as ABIS individuals without any future autoimmune disease ($n = 15,732$).

Proteomic profiling of cord blood from 1,204 children is performed (Fig. 1b). As a number of these children had other diagnoses in this period, their samples are used here only for environmental association. Machine learning (ML) models identify predictive proteins in 432 selected children, including controls and cases with future T1D (Fig. 1c). Non-linear protein-metabolite-exogenous compound associations are determined using Shapley Additive exPlanations[24] (SHAP) in 132 children with metabolomic data (Fig. 1d) and associations with environmental factors assessed, irrespective of future disease. Several key risk factors and proteins are significantly relevant to future disease on the basis of these approaches (Fig. 1e).

### Prenatal risk factors associated with future T1D

Birth questionnaires are assessed for association with future T1D diagnosis, comparing 15,732 controls to 167 cases. A total of 44 variables are tested, including perinatal factors, family medical history, parent factors, prenatal stressors, prenatal diet, and medications and infections during pregnancy (Supplementary Data 1). The most significant birth factors are those related to family medical history of (e.g., T1D, T2D, and asthma). However, also significant are (1) perinatal factors (e.g., caesarean section, placement in a newborn care unit); (2) demographics; (3) stomach flu during pregnancy; (4) medications taken during pregnancy (e.g., antibiotics and psychotropics); whether time is spent with someone with insulin-dependent diabetes during pregnancy; and to a limited extent, prenatal diet (Fig. 2a–c and Supplementary Data 1).

These factors are carried into ML models for prediction. With the ABIS cohort having the expected global T1D incidence of 1%, class-weighted loss functions are employed to mitigate the effect on performance and generalizability. However, the predictive performance of logistic regression, eXtreme Gradient Boosting (XGBoost), Random Forest, and Support Vector Machine (SVM) models using these factors alone is limited, with area under the receiver operating characteristic curve (AUC-ROC) values from 0.43 to 0.62 (Fig. 2d).

### Cord blood proteomic analyses

Selected inflammatory and immune markers are analyzed from cord blood samples spanning 146 cases and 286 controls, measured using OLINK Explore 384 Inflammation 1 and 2 panels and the Target Immune Response panel. Controls are defined as those individuals having no future autoimmune diagnoses, psychiatric conditions, or neurodevelopmental disorders, while cases had confirmed T1D diagnoses (up to 22 years of age). Controls are selected at random from the larger ABIS study. Given the significant class imbalance described above, the minority and majority classes are intentionally oversampled and undersampled, respectively.

Consistent with the full ABIS study, individuals in the T1D proteomic group are significantly more likely to have been placed in a pediatric ward as newborns (odds ratio, OR = 2.1, $p = 0.015$), with 13.9% of the T1D group experiencing this compared to 6.6% of controls. As expected, there is a higher likelihood of T1D among family members, including the mother ($p < 0.001$), father ($p = 0.003$), and grandparents ($p = 0.014$). Specifically, 29.5% of T1D individuals had a family history of T1D compared to only 10.5% of controls ($p < 0.00001$; OR = 3.6, 95% CI: 2.1–6.0, $p < 0.0001$). No significant differences are observed between groups in terms of biological sex, mode of delivery, gestational age, birth weight, maternal BMI (both pre- and post-pregnancy), parental age at birth, or other assessed environmental factors, including infection or smoking exposure during pregnancy and severe life events (Supplementary Data 2).

Groups are broadly representative relative to the overall ABIS cohort, showing no major evidence of selection bias (Table 1 and Supplementary Data 3). The only notable difference is a modest overrepresentation of females among controls included in the Olink subset (52.5%) versus those controls who are not (47.9%). Gestational age is slightly higher in T1D subjects selected for Olink analysis (39.7 ± 1.6 versus 38.4 ± 3.1 weeks). Nevertheless, the selection of controls seen in the Olink case/control cohort is sufficient to ensure that the Olink groups are broadly representative of the full ABIS study, with no major evidence of selection bias (Table 1). NPX levels are assessed across the entire case/control proteomics cohort (between future T1D, $n = 146$, and controls, $n = 286$; Fig. 3) and then stratified by HLA genetic risk for T1D (see below).

### Global protein concentration differences by future diagnosis

Across the 386 proteins assessed (Supplementary Data 4), 32 are significantly, differentially expressed after false discovery rate (FDR) correction (Fig. 3a and Supplementary Data 5 and 6). Four proteins are significantly higher in cases: HLA class II histocompatibility antigen, DR alpha chain (HLA-DRA; $p = 8.98e-19$, $q = 6.53e-16$); iduronate 2-sulfatase (IDS; $p = 6.65e-13$, $q = 2.42e-10$); secretoglobin family 3 A member 2 (SGB3A2; $p = 0.00031$, $q = 0.014$), and cathepsin C (CTSC; $p = 0.00040$, $q = 0.017$).

Among the 28 proteins significantly higher in controls after FDR correction were: (1) tissue inhibitor of metalloproteinases 3 (TIMP3; $p = 3.68e-9$, $q = 8.91e-7$); CD40 ligand (CD40LG; p = 1.44e-7, $q = 2.61e-5$); matrix extracellular phosphoglycoprotein (MEPE; $p = 1.09e-6$, $q = 1.58e-4$); adenosine deaminase (ADA; $p = 2.51e-6$, $q = 2.98e-4$); neurotrophin-3 (NTF3; $p = 2.87e-6$, $q = 2.98e-4$); CD84 molecule (CD84; $p = 5.34e-6$, $q = 4.85e-4$); serine protease inhibitor Kazal-type 2 (SPINT2; p = 1.28e-5, $q = 0.0010$); PDZ and LIM domain protein 7 (PDLIM7; $p = 7.72e-5$, $q = 0.0056$); and linker for activation of T cells (LAT; $p = 8.49e-5$, $q = 0.0056$). Strong correlations in proteins elevated in controls are observed: lymphocyte-specific protein 1 (LSP1) and ADA ($R = 0.59$), tissue inhibitor of metalloproteinases 3 (TIMP3) and CD40LG ($R = 0.5$), and PDLIM7 and TBC1 domain family member 5 (TBC1D, $R = 0.79$, $p$'s < 2.2e-16).

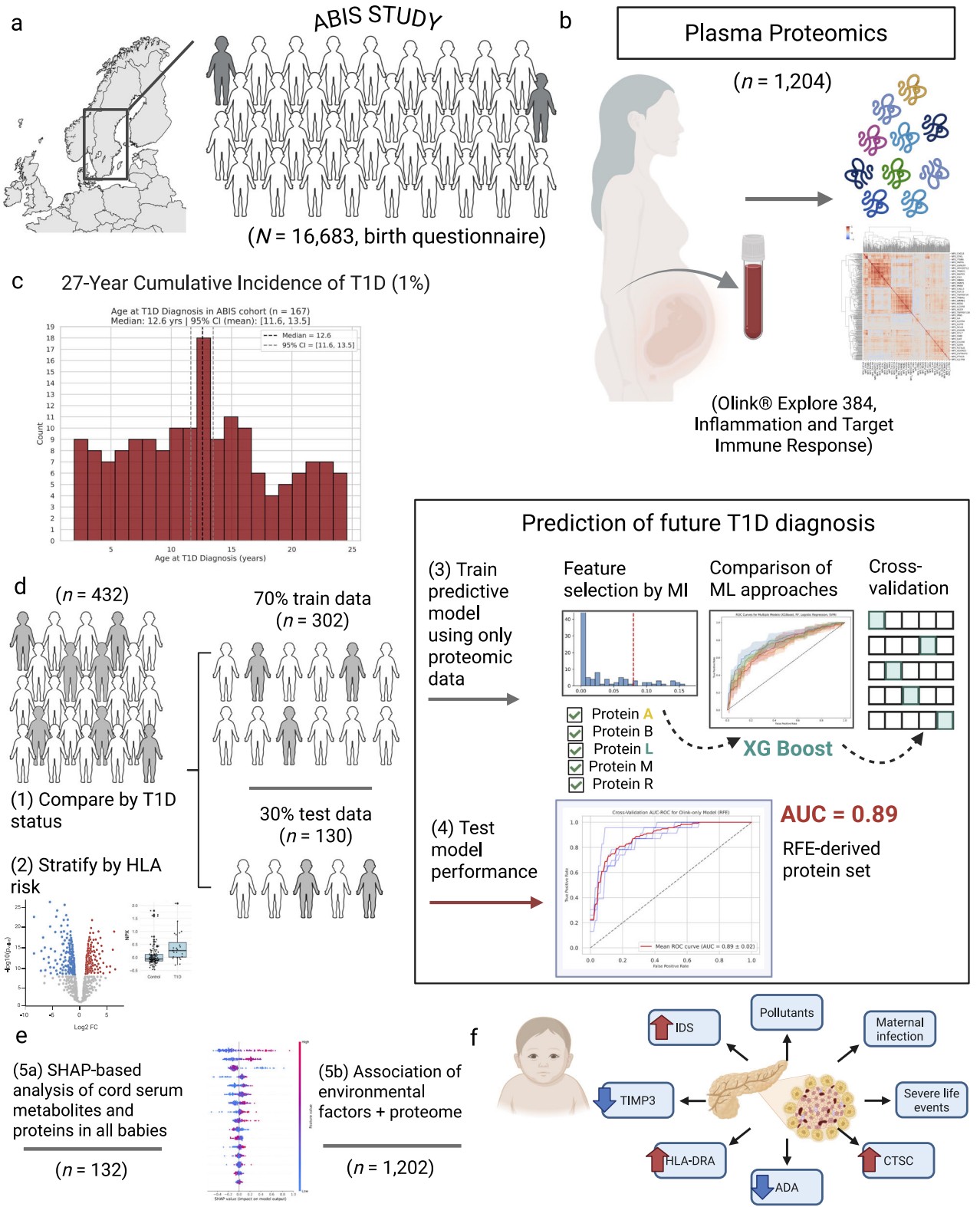

To ensure balanced comparison, a 1:1 selection of cases and controls is performed using propensity score matching ($n = 280$), accounting for significant factors including family history of T1D (in the mother), mode of delivery, sex of the child, week of delivery, vulnerability index, serious life events, and stomach flu during pregnancy. Subsequent analysis, adjusted for confounders and corrected for multiple comparisons, confirmed the same markers identified in the global analysis, demonstrating that the observed effects persist independently of family history (Fig. 2e). Notably, serine protease prostasin (PRSS8) also emerged as significant.

## Gene enrichment of T1D-associated proteins
Gene Set Enrichment Analysis (GSEA) showed that the T1D-associated proteins serve as core components in pathways involving cytokine signaling and immune responses. Involved in cytokine pathways are HLA-DRA, CD40LG, and TIMP3, while others, such as CTSC, linker for

**Fig. 1 | Analytical framework for type 1 diabetes (T1D) risk in the ABIS Study.** **a** Overview of the All Babies in Southeast Sweden (ABIS) study, which enrolled families between 1997 and 1998 at obstetric clinics in a six-county region of Sweden. Over 27 years of follow-up, the cumulative incidence of T1D is 1%. **b** Proteomic profiling of cord blood from 1202 children is performed using Olink Explore 384, Inflammation 1 and 2 panels, and the Target Immune Response panel. The full dataset is analyzed to assess relationships between the top proteins (identified in the case/control analysis) and birth-related factors. **c** Olink analysis aimed at identifying group differences included 432 children (future T1D cases and controls). Data are analyzed both globally and stratified by HLA risk groups using Wilcoxon tests with false discovery rate (FDR) correction. Machine learning (ML) models are developed, with eXtreme Gradient Boosting (XGBoost) as a base estimator on recursive feature elimination (RFE)-selected proteins achieving the highest accuracy after fivefold cross-validation. **d** SHapley Additive exPlanations (SHAP)-based analysis of the most predictive proteins revealed non-linear associations between proteins and metabolites in 132 children with available metabolomic data. **e** An integrated model highlights key risk factors and proteins associated with T1D pathogenesis, hypothesized to contribute to beta-cell stress and damage, ultimately leading to T1D. **f** T1D diagnoses in the ABIS cohort, depicting the median and 95% confidence interval of age at time of diagnosis. MI mutual information, IDS iduronate 2-sulfatase, TIMP3 tissue inhibitor of metalloproteinases 3, HLA-DRA, human leukocyte antigen DR alpha chain, ADA adenosine deaminase, CTSC cathepsin C. Created in BioRender. Ahrens, A. (2025) https://BioRender.com/k05i837.

activation of T cells (LAT), CD40LG, CD84, and ADA, play central roles in pathways related to immune system regulation, leukocyte function, the Reactome, and both adaptive and innate immune signaling (Fig. 3c–e). In the future T1D group, two gene sets, SWEET_LUNG_CANCER_KRAS_UP and GOCC_CELL_SURFACE, are upregulated. In contrast, nine gene sets are downregulated, reflecting disruptions in stress response, hydrolase regulation, and intracellular signaling pathways (Supplementary Data 7). The enriched gene sets are associated with notable core proteins: (1) TIMP3 and neurotrophin-3 (NTF3) with six of the nine downregulated gene sets; (2) forkhead box protein O1 (FOXO1) with five; (3) caspase-2 (CASP2) and MAP2K6 (mitogen-activated protein kinase kinase) with four (Fig. 3c); and 4) HLA-DRA shared between the two upregulated gene sets.

## Protein concentration differences by age at diagnosis
Future cases are categorized by diagnosis at ≤5 years, 6–10 years, 11–17 years, and 18–24 years. Each group is compared to controls (Fig. 4a–c and Supplementary Fig. 1a), with results in Supplementary Data 8. The most significant proteomic differences are observed among individuals diagnosed by age 5, with 114 significant proteins, 64 remaining significant after FDR. Shared across at least three age groups are reduced ADA, TIMP3, CD40LG, and signaling threshold-regulating transmembrane adaptor 1 (SIT1), alongside increased HLA-DRA, IDS, secretoglobin family 3 A member 2 (SCGB3A2), and CTSC. Unique to newborns who developed T1D in adulthood (18–24 years) are increased interleukin-20 (IL20) and C-C motif chemokine ligand 4 (CCL4)—also called macrophage inflammatory protein-1β, MIP-1β—and decreased Wnt family member 9 A (WNT9A, Supplementary Fig. 1a).

Protein-protein association network and functional pathway enrichment analyses[25] of the FDR-corrected protein differences in earliest-diagnosed cases (by age five) demonstrate significant interaction enrichment (protein-protein interaction (PPI) enrichment, $p = 2.03e-07$, Supplementary Fig. 1b). Significant enrichments of pathways related to toxoplasmosis; positive regulation of protein serine/threonine kinase activity, transferase activity, and MAP kinase activity; modulation of T-cell activation and inflammatory responses; vitamin D in inflammatory disease; *Yersinia* infection; signaling of RANKL/RANK, C-type lectin receptors, T-cell receptors, NF-κB, Ebstein-Barr virus LMP1, and cell surface receptors wer found—all significant after FDR (Fig. 4d–f and Supplementary Data 9).

## Protein concentration differences stratified by HLA genetic risk
Next, future T1D cases ($n = 146$) are compared to two subsets of controls (81 controls with genetic risk for T1D and 76 controls without risk; Supplementary Data 10). HLA risk is defined by the presence MHC Class II alleles DR4-DQ8 (DRB1-DQA103:01-DQB103:02) and/or DR3-DQ2 (HLA-DRB103:01-DQA105:01-DQB1*02:01)[23,26], seen in as many as 90% of T1D patients[27,28], as these represent the primary risk factor for islet autoimmunity[29,30].

Comparing non-genetic risk controls (lacking DR4-DQ8 or DR3-DQ2 alleles) versus future T1D (Fig. 5a), four proteins are significant after FDR correction: HLA-DRA and IDS (higher in T1D) and TIMP3 and NTF3 (higher in low-risk controls (Fig. 5a). Consistent with global T1D results, the greatest differences are in HLA-DRA and IDS (higher in T1D) and TIMP3 and NTF3 (higher in controls without DR4-DQ8 or DR3-DQ2).

Next, future T1D cases are separated by HLA risk (decreased/neutral, increased/high) and compared to controls with similar genetics (Supplementary Data 10). Decreased/neutral HLA risk genotypes are seen in 125 controls and 18 with future T1D, with 20 significant proteins significant, with persistent higher levels of IDS (Fig. 5b). HLA-DRA is also higher in future T1D without HLA genetic risk, as is seen in global T1D comparisons (Fig. 5b), but not significant after FDR, as are CCL4, C-C motif chemokine ligand 21 (CCL21), lymphocyte antigen 9 (LY9, also called SLAMF3), lymphocyte antigen 75 (LY75, also called DEC-205 or CD205), tumor necrosis factor (TNF) receptor superfamily member 13B (TNFRSF13B, also called TACI), CD70 molecule (also called TNF ligand superfamily member 7, TNFSF7) CD70, signal regulatory protein beta 1 (SIRPB1), and Erb-B2 receptor tyrosine kinase 3 (ERBB3). Notably, TIMP3 is not significant between cases and controls both lacking high-risk HLA for T1D.

Far more differences are seen when comparing those with genetic risk, even after FDR adjustment (Fig. 5c, d). When comparing all future T1D to controls with either DR4-DQ8 or DR3-DQ2, four differential proteins are identified, after FDR, including SCGB3A2, HLA-DRA, and IDS (higher in T1D), as well as TIMP3 (higher in controls) (Fig. 5c). Next, subsets of children with high/increased risk are compared ($n = 76$ T1D, $n = 44$ controls). Eleven proteins are significantly different after FDR (Fig. 5d), including: HLA-DRA, IDS, and SCGB3A2 (higher in high-risk T1D) and TIMP3, LAT, CD40LG, ADA, PDLIM7, integral membrane protein 2 A (ITM2A), and CD84 (higher in high-risk controls). In individuals lacking DR3-DQ2 ($n = 53$ T1D, $n = 112$ controls), with most T1D cases carrying DR4-DQ8, numerous differences are observed (Fig. 6a, b). The proteins most enriched in T1D after FDR are HLA-DRA, IDS, and IL20. In contrast, proteins like ADA, aldehyde dehydrogenase 3 family member A1 (ALDH3A1), CD40LG, LSP1, MEPE, NTF3, and TIMP3 are more highly expressed in controls. Conversely, in the subset lacking DR4-DQ8 ($n = 24$ T1D, $n = 111$ controls), fewer significant markers are identified, not reaching significance after FDR correction (Supplementary Fig. 2a, b).

## Prediction of future T1D using proteomics at birth
By prioritizing proteins with high mutual information (MI) scores, T1D prediction models are focused on proteins with the highest relative influence on prediction, capturing meaningful and potentially complex protein relationships. The mean relative concentration levels of the top 40 proteins (Supplementary Fig. 3a), and their relative contribution to case and control distinction (Supplementary Fig. 3b), are assessed. Key contributors tested are HLA-DRA, IL16, PTH1R, GAL, and IDS. The observed differences between future T1D and controls aligned with previously identified markers such as HLA-DRA, IDS, PDLIM7, SPINT2, ADA, TIMP3, and LAT (Supplementary Fig. 3c).

Diagnostic accuracy of XGBoost, Random Forest, Logistic Regression, and SVM models is assessed using the top 40

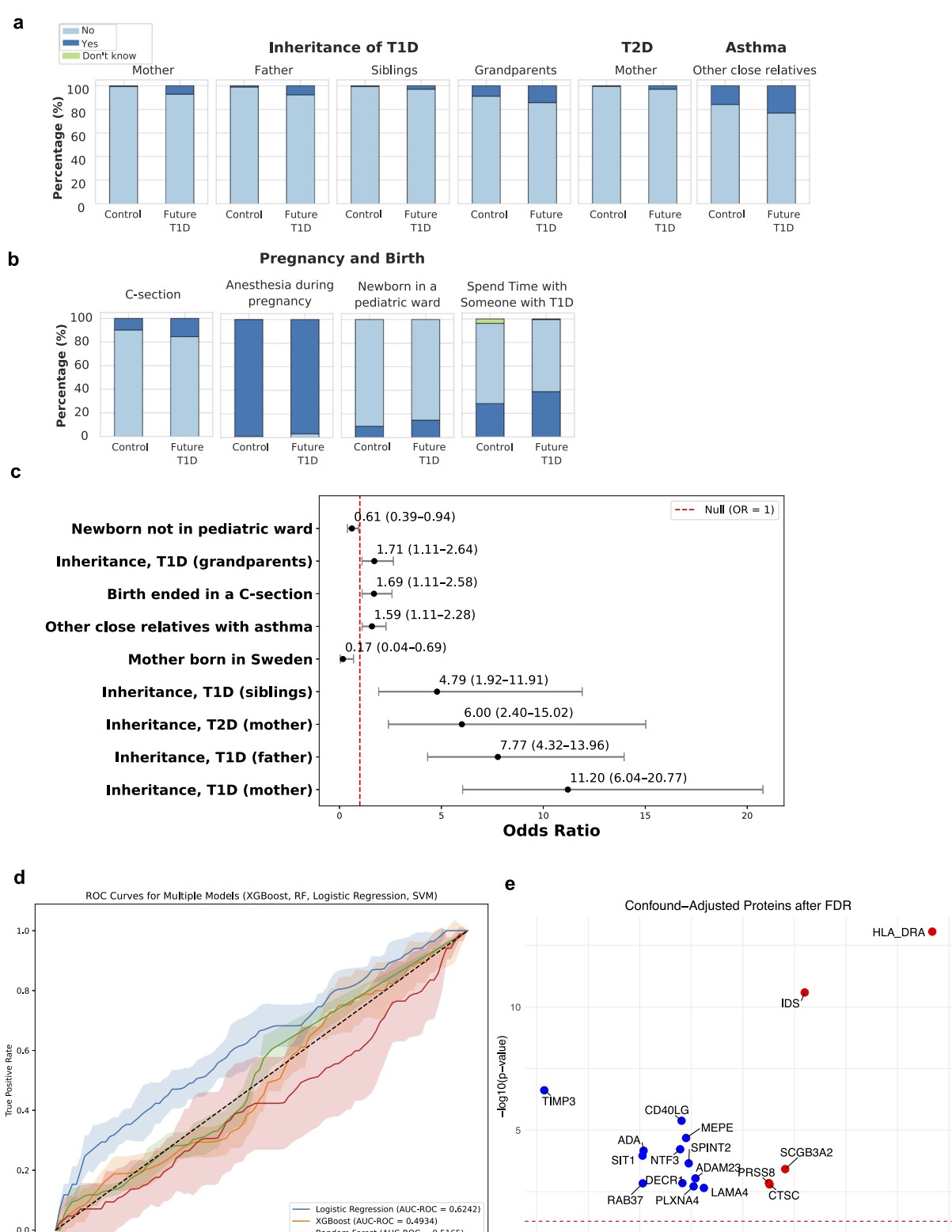

(Supplementary Fig. 3d) and a refined set of 32 proteins (via the elbow method; Supplementary Fig. 3e), to predict future T1D using MI scoring. While Random Forest slightly outperformed in AUC-ROC, XGBoost matched closely and delivered a more robust F1 score when using 40 proteins, making it the preferred model for prediction. With fivefold cross-validation, XGBoost achieved an AUC-ROC of $0.77 \pm 0.04$, with precision ($0.63 \pm 0.05$), recall ($0.69 \pm 0.07$), and a

strong F1 score ($0.65 \pm 0.04$), all showing that the model is generalizable and not overfitting. Prediction of future T1D, using these proteins alone, achieved high accuracy (AUC = 0.77).

Given the established association of DR4-DQ8 with early-onset T1D[31,32], we apply a separate predictive model in a restricted subset of children, comparing controls to individuals who later developed T1D but also carry DR4-DQ8 alleles (Supplementary Fig. 3f and

**Fig. 2 | Prenatal factors associated with future type 1 diabetes (T1D) diagnosis in ABIS. a** Distribution of family history of disease and **b** pregnancy and birth factors significantly associated with future T1D diagnosis, based on 167 T1D cases and 15,543 controls from the full All Babies in Southeast Sweden (ABIS) study. Sample sizes for cases and controls are provided in Supplementary Data 1 for each factor. **c** Odds ratios and 95% confidence intervals for dichotomous variables (i.e., with 1 degree of freedom). Statistics are provided in Supplementary Data 1, with controls as the reference group. **d** Receiver operating characteristic (ROC) curve analysis comparing the accuracy of traditional and machine learning models, including logistic regression, eXtreme Gradient Boosting (XGBoost), Random Forest, and Support Vector Machine (SVM) using these factors alone, highlighting suboptimal predictive performance. Supplementary Data 1 provides chi-square statistics and the case/control counts for parent-reported variables on the birth questionnaire in the full ABIS cohort. **e** Proteins differentiating future T1D and controls, after controlling for significant environmental factors by propensity score matching in a 1:1 ratio using nearest neighbor (n = 280). Only the proteins that are significant after FDR correction for multiple comparisons are indicated, with median fold change >0 indicating proteins higher in abundance in T1D and <0 indicating those higher in matched controls. ADAM23 disintegrin and metalloproteinase domain-containing protein 23, ADA adenosine deaminase, C-section cesarean section, CD40LG CD40 ligand, CTSC cathepsin C, DECR1 2,4-dienoyl-CoA reductase 1, FDR false discovery rate, HLA-DRA histocompatibility antigen, DR alpha chain, IDS iduronate 2-sulfatase, LAMA4 laminin subunit alpha-4, MEPE matrix extracellular phosphoglycoprotein, NTF3 neurotrophin-3, OR odds ratio, PLXNA4 plexin A4, PRSS8 serine protease 8, SCGB3A2 secretoglobin family 3A member 2, SIT1 signaling threshold-regulating transmembrane adaptor 1, SPINT2 serine protease inhibitor, Kunitz type 2, T2D type 2 diabetes, TIMP3 tissue inhibitor of metalloproteinases 3, XGBoost eXtreme Gradient Boosting.

## Table 1 | Comparison of cohort characteristics by diagnostic group and inclusion in the proteomic analysis

| | | Controls | | | Future T1D | | |
| --- | --- | --- | --- | --- | --- | --- | --- |
| | | Controls—no Olink | Controls—Olink | | T1D—no Olink | T1D—Olink | |
| | | Count (%) or Mean | Count (%) | p | Count (%) or Mean | Count (%) | p |
| **Sex** | Male | 7889 (52.1%) | 503 (47.5%) | 0.004 | 12 (57.1%) | 81 (55.5%) | 0.886 |
| | Female | 7251 (47.9%) | 555 (52.5%) | | 9 (42.9%) | 65 (44.5%) | |
| **The birth ended with a caesarean section** | No | 13669 (90.1%) | 961 (90.8%) | 0.447 | 16 (76.2%) | 125 (85.6%) | 0.265 |
| | Yes | 1500 (9.9%) | 97 (9.2%) | | 5 (23.8%) | 21 (14.4%) | |
| **Week of delivery** | | 39.7 ± 1.8 | 39.8 ± 1.7 | 0.501 | 38.4 ± 3.1 | 39.7 ± 1.6 | 0.118 |
| **Inheritance, T1D (mother)** | No | 15062 (99.3%) | 1048 (99.1%) | 0.373 | 19 (90.5%) | 136 (93.2%) | 0.657 |
| | Yes | 107 (0.7%) | 10 (0.9%) | | 2 (9.5%) | 10 (6.8%) | |
| **Inheritance, T1D (father)** | No | 15009 (98.9%) | 1043 (98.6%) | 0.269 | 20 (95.2%) | 134 (91.8%) | 0.58 |
| | Yes | 160 (1.1%) | 15 (1.4%) | | 1 (4.8%) | 12 (8.2%) | |
| **Smoking during 30–32 pregnancy weeks** | non-smoker | 4139 (90.2%) | 255 (91.4%) | 0.603 | 6 (85.7%) | 38 (95%) | 0.354 |
| | 1–9 cig/day | 350 (7.6%) | 17 (6.1%) | | 1 (14.3%) | 2 (5%) | |
| | 10+ cig/day | 99 (2.2%) | 7 (2.5%) | | 0 (0%) | 0 (0%) | |
| **Vulnerability index score** | 0 = lowest | 5830 (41.2%) | 405 (40.9%) | 0.345 | 6 (33.3%) | 70 (49.3%) | 0.598 |
| | 1 | 5450 (38.5%) | 362 (36.6%) | | 8 (44.4%) | 52 (36.6%) | |
| | 2 | 1937 (13.7%) | 151 (15.3%) | | 3 (16.7%) | 14 (9.9%) | |
| | 3 = highest | 926 (6.5%) | 72 (7.3%) | | 1 (5.6%) | 6 (4.2%) | |
| **Did you smoke during your pregnancy?** | No | 13132 (88.8%) | 918 (88.9%) | 0.977 | 17 (89.5%) | 131 (91.6%) | 0.756 |
| | Yes | 1650 (11.2%) | 115 (11.1%) | | 2 (10.5%) | 12 (8.4%) | |
| **Severe life event during pregnancy** | Yes | 1379 (9.4%) | 91 (8.9%) | 0.608 | 2 (11.1%) | 14 (9.7%) | 0.852 |
| | No | 13355 (90.6%) | 934 (91.1%) | | 16 (88.9%) | 130 (90.3%) | |
| **Did the mother have stomach flu during pregnancy?** | Yes | 4925 (33.6%) | 323 (31.4%) | 0.273 | 11 (57.9%) | 56 (38.9%) | 0.267 |
| | No | 9395 (64.1%) | 678 (65.9%) | | 8 (42.1%) | 86 (59.7%) | |
| | Do not know | 338 (2.3%) | 28 (2.7%) | | 0 (0%) | 2 (1.4%) | |
| **Education of mother** | Low | 1274 (8.6%) | 89 (8.6%) | 0.496 | 2 (10.5%) | 13 (9%) | 0.374 |
| | Medium | 8783 (59.4%) | 632 (61.2%) | | 15 (78.9%) | 95 (66%) | |
| | High | 4719 (31.9%) | 312 (30.2%) | | 2 (10.5%) | 36 (25%) | |
| **Education of father** | Low | 1965 (13.5%) | 147 (14.5%) | 0.319 | 5 (26.3%) | 25 (17.5%) | 0.105 |
| | Medium | 8996 (61.8%) | 638 (62.8%) | | 13 (68.4%) | 79 (55.2%) | |
| | High | 3594 (24.7%) | 231 (22.7%) | | 1 (5.3%) | 39 (27.3%) | |

To evaluate whether the subset with Olink proteomic data is representative of the full cohort, cohort characteristics were compared between participants included and not included in the Olink analysis, stratified by T1D outcome. Chi-square and Wilcoxon rank-sum tests were used where appropriate. See also Supplementary Data 3.

Supplementary Fig. 4). Following hyperparameter tuning via Grid-Search, the model achieves a mean AUC-ROC of 0.83 ± 0.12 (Supplementary Fig. 3g), further increasing prediction accuracy. It demonstrated robust performance across classification metrics, including an F1 score of 0.62 ± 0.16, precision of 0.66 ± 0.20, recall of 0.63 ± 0.20, and an overall accuracy of 0.84 ± 0.07.

Finally, we apply a supervised ML approach using recursive feature elimination (RFE) on the full Olink case/control dataset (n = 432). The data are split into training (80%) and testing (20%) sets, preserving class proportions. RFE is applied to recursively select the top 30 most predictive features, first excluding (Fig. 7a, b) and then including (Fig. 7c, d) dosage of HLA high-risk alleles DR4DQ8 and DR3DQ2. We

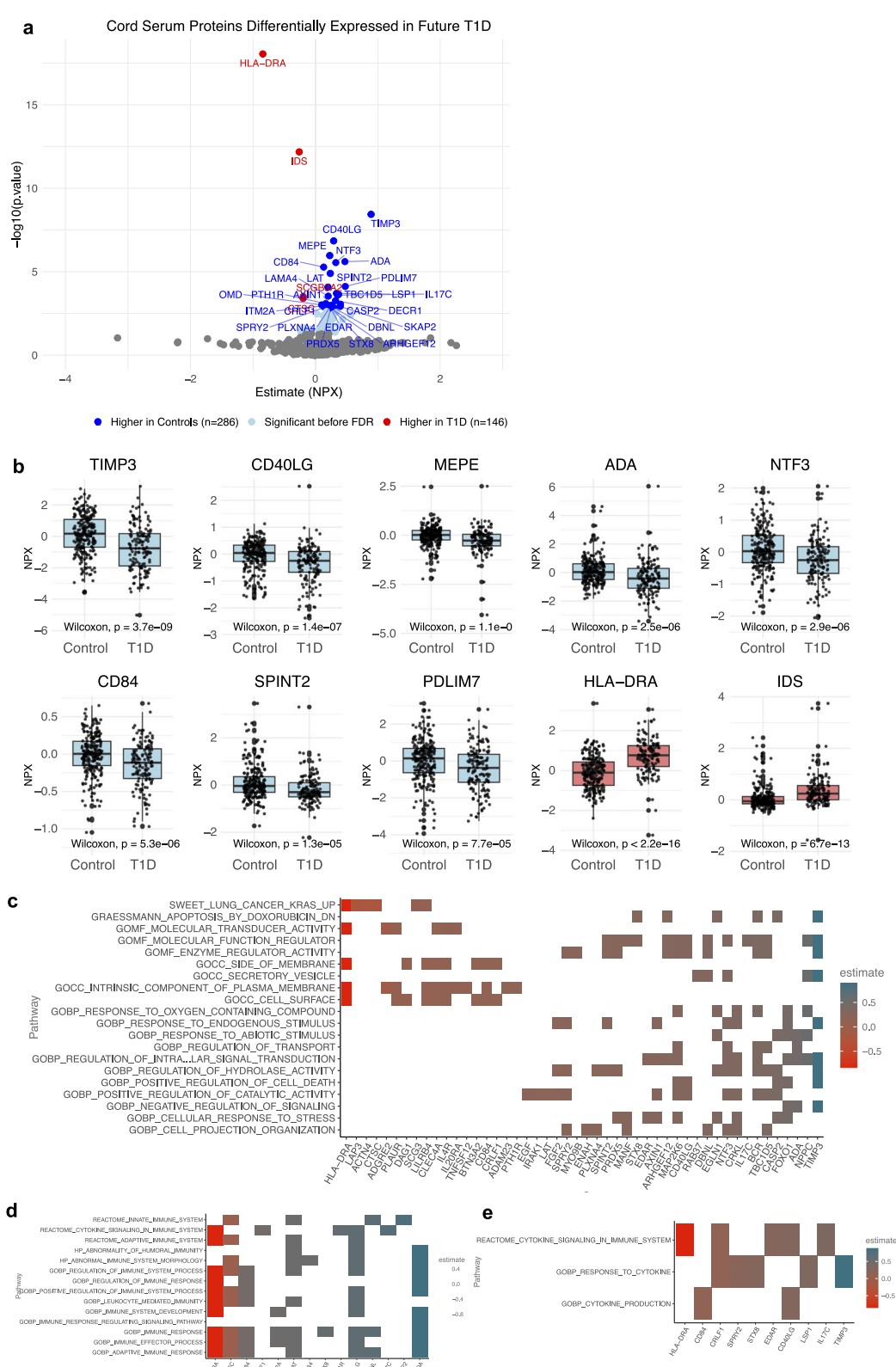

perform fivefold stratified cross-validation using these fixed selected features within each fold to evaluate generalizability and maintain interpretability. Both models, Olink-only and Olink + HLA, perform similarly. The AUC-ROC is 0.89 ± 0.02 without HLA dosage (Fig. 7a) and 0.89 ± 0.03 with HLA dosage (Fig. 7c), indicating no improvement from adding HLA genetic information. The Olink + HLA model yields an F1 score of 0.70 ± 0.06, precision of 0.76 ± 0.10, and recall of

0.67 ± 0.08. The Olink-only model has slightly better performance by its F1 score of 0.73 ± 0.05 and recall of 0.71 ± 0.07, but precision of 0.74 ± 0.06. Feature selection by RFE demonstrates highly overlapping predictive protein sets, with 27 of the same 29 proteins (93%) shared between models (Fig. 7b, d). HLA-DRA and IDS are consistently ranked the top two proteins. The median difference in protein abundance in the RFE-selected proteins for the Olink + HLA model is presented for

**Fig. 3 | Significant differences in normalized protein expression (NPX) in infants with future type 1 diabetes (T1D). a** Proteins differentiating future T1D ($n = 146$) and controls ($n = 286$), based on log2fold change statistics. Controls are defined as children without a future diagnosis of an autoimmune disease, psychiatric condition, or neurodevelopmental disorder. Annotated proteins are significant after false discovery rate (FDR) correction. **b** Wilcoxon statistics with a two-sided test for group differences (controls and future T1D) for each of the top ten proteins. For means, see Supplementary Data 4 and for full statistics, see Supplementary Data 5. Boxplots show the median (line) and interquartile range (box, 25th–75th percentile) to demonstrate the data distribution, future T1D ($n = 146$) and controls ($n = 286$). Whiskers extend to the most extreme values within 1.5 × interquartile range (IQR) of the lower and upper quartiles; points beyond the whiskers are plotted as outliers. Values are displayed as normalized protein expression (NPX) values. **c** T1D-associated proteins identified as core proteins in Gene Set Enrichment Analysis (GSEA) pathways, and those specifically related to: **d** cytokines and **e** the immune system. ADA adenosine deaminase, ARHGEF12 Rho guanine nucleotide exchange factor 12, CASP2 caspase-2, CD40LG CD40 ligand, CD84 CD84 molecule, DBNL drebrin-like protein, DECR1 2,4-dienoyl-CoA reductase 1, EDAR ectodysplasin A receptor, GOBP Gene Ontology Biological Process, GOCC Gene Ontology Cellular Component, GOMF Gene Ontology Molecular Function, HLA-DRA histocompatibility antigen, DR alpha chain, IDS iduronate 2-sulfatase, IL17C interleukin-17C, ITM2A integral membrane protein 2A, LAMA4 laminin subunit alpha-4, LAP3 leucine aminopeptidase 3, LSP1 lymphocyte-specific protein 1, MEPE matrix extracellular phosphoglycoprotein, NTF3 neurotrophin-3, OMD osteomodulin, PDLIM7 PDZ and LIM domain protein 7, PLAUR plasminogen activator urokinase receptor, PLXNA4 plexin A4, PRDX5 peroxiredoxin-5, PTH1R parathyroid hormone 1 receptor, SKAP2 src kinase-associated phosphoprotein 2, SPINT2 serine protease inhibitor, Kunitz type 2, SPRY2 sprouty RTK signaling antagonist 2, STX8 syntaxin-8, TBC1D5 TBC1 domain family member 5, TIMP3 tissue inhibitor of metalloproteinases 3.

cases and controls in Fig. 7e. Given the similar predictive value of the models, to explore redundancy between allele dosage and HLA-DRA protein levels, an ANOVA test compares HLA-DRA abundance by dosage groups, revealing a strong association ($p = 1.12e-08$, Fig. 7f).

## Metabolites, exogenous toxic compounds (PFOS), and T1D-linked proteins

Using non-parametric correlation methods in the 132 individuals with both metabolomic and proteomic data, several proteins enriched in controls—TIMP3 and PDLIM7—are significantly associated with cord serum metabolites. PDLIM7 showed positive correlations with multiple metabolites, including aspartic acid, deoxynivalenol (DON), fumaric acid, lysophospholipid lysophosphatidylethanolamine (lysoPE) 18:0, and malic acid (FDR-adjusted $p = 0.0055$–$0.032$). TIMP3 shows broader associations, including positive correlations with 3-indoleacetic acid, mycotoxin DON, glutamic acid, isocaproic acid, isoleucine/leucine, isovaleric acid, lysine, proline, and tyrosine (FDR-adjusted $p = 0.012$–$0.042$). Although nonparametric and less sensitive to extremes, the Spearman correlation is not entirely immune to outliers, which can still affect the rank order. To address this concern, we repeated the analysis removing the most extreme 5% of samples (lowest 2.5% and highest 2.5%), and results are largely consistent for TIMP3 (Supplementary Fig. 5; FDR-adjusted $p$'s = 0.006–0.03), suggesting that a small number of influential outliers did not sway the results. However, for PDLIM7, only lysoPE 18:0 remains significant (rho = 0.36, FDR-adjusted $p = 0.007$), along with perfluorinated compound perfluorooctanoic acid, branched isomer (PFOAbr, rho = 0.32, FDR-adjusted $p = 0.03$).

Given that perfluorooctane sulfonate (PFOS) is a persistent environmental contaminant with immunomodulatory and metabolic effects, we next examine the correlation of circulating proteins with PFOS-L, the linear isomer that is often the dominant form in environmental samples. Using a nonparametric Spearman test, we observe directional associations of PFOS with the following immune proteins: IL18 ($p = 0.003$), SELPLG ($p = 0.016$), TNFRSF4 ($p = 0.027$), DAPP1 ($p = 0.027$), IL20 ($p = 0.042$), COLEC12 ($p = 0.043$), and ESM1 ($p = 0.048$) show negative correlations, while IL17A ($p = 0.020$) and ENPP5 ($p = 0.038$) are positively correlated (Fig. 8). Although these trends do not survive FDR correction due to the smaller sample size ($n = 132$), they highlight biologically plausible immune pathways that may be influenced by PFOS exposure for future study.

To complement this univariate analyses, SHAP (Shapley Additive exPlanations[24]) is applied to assess multivariate, non-linear associations between metabolites and exogenous compounds with the five proteins most predictive of future T1D: HLA-DRA, IDS, and CTSC (elevated in T1D), and ADA and TIMP3 (elevated in controls), as identified through both traditional (Figs. 3 and 4) and ML approaches (Fig. 7 and Supplementary Fig. 3). SHAP values reveal complex feature interactions and non-linear patterns for each protein (Fig. 9a), with the top metabolites identified by MI score (Fig. 9a–f). TIMP3 concentration is positively associated with lysine and tauro-β-muricholic acid (TbMCA) while negatively associated wtih stearic acid (Fig. 9b). For ADA concentration, arachidic acid and PFOS-L are positively and negatively associated, respectively (Fig. 9c). Stearic acid consistently demonstrates a strong positive association with IDS and HLA-DRA concentration (Fig. 6d, e), as does PFOS-L with HLA-DRA, although clustering suggests non-linearity. Examples of negative contributors to IDS and HLA-DRA concentration include LPE-16:0 and pyroglutamic acid, respectively (Fig. 9f).

## Prenatal/perinatal factors and T1D-associated proteins

In the full Olink dataset ($n = 1202$), associations are discovered between the 65 proteins most significant across T1D and control global comparisons (from Fig. 3). Wilcoxon statistics are run across these proteins with respect to biological sex and several perinatal and prenatal factors on the birth questionnaires (Fig. 9g, h and Supplementary Data 11). After FDR correction, three proteins higher in males, corneodesmosin (CDSN), SH2 domain–containing protein 1 A (SH2D1A), and cathepsin O (CTSO), remained significantly associated (Fig. 9g), and another 24 proteins are associated with mode of delivery (Fig. 9h). Protein associations with other environmental factors are found, including smoking of the mother, maternal stomach flu, and severe life events during pregnancy, but none are significant after FDR and thus are not presented.

## Discussion

Although T1D is manageable with insulin, it requires meticulously maintaining proper blood glucose levels[33]. Life-threatening complications are omnipresent. There remains a critical need for early screening methods to predict future T1D prior to pancreatic β-cell destruction. There is currently little guidance surrounding metabolomic screening for pre-clinical T1D[34]. This study offers necessary, preliminary evidence that proteomic markers present at birth could serve as valuable predictors in a general population, paving the groundwork for accessible, cost-effective newborn screening tools. The proteins discovered in our study effectively differentiate children who develop T1D from those who do not, particularly in the first five years of life. This proteomics-based approach could complement current practices, allowing more refined and timely identification of children at risk of T1D for follow-up autoantibody monitoring, based on these conditions, which we hypothesize are prime for β-cell destruction prior to autoimmunity. Mechanisms surrounding tissue repair/remodeling, extracellular matrix (ECM) modeling, and immune activation are shared in the network of these proteins, based on GSEA and STRING analysis. High HLA-DRA could reflect an overactive antigen presentation at birth, even in children without significant T1D genetic risk. HLA-DRA is involved in presenting peptides derived from pathogen-derived peptides to T cells. In islet cells, IDS enhances glucose-induced insulin

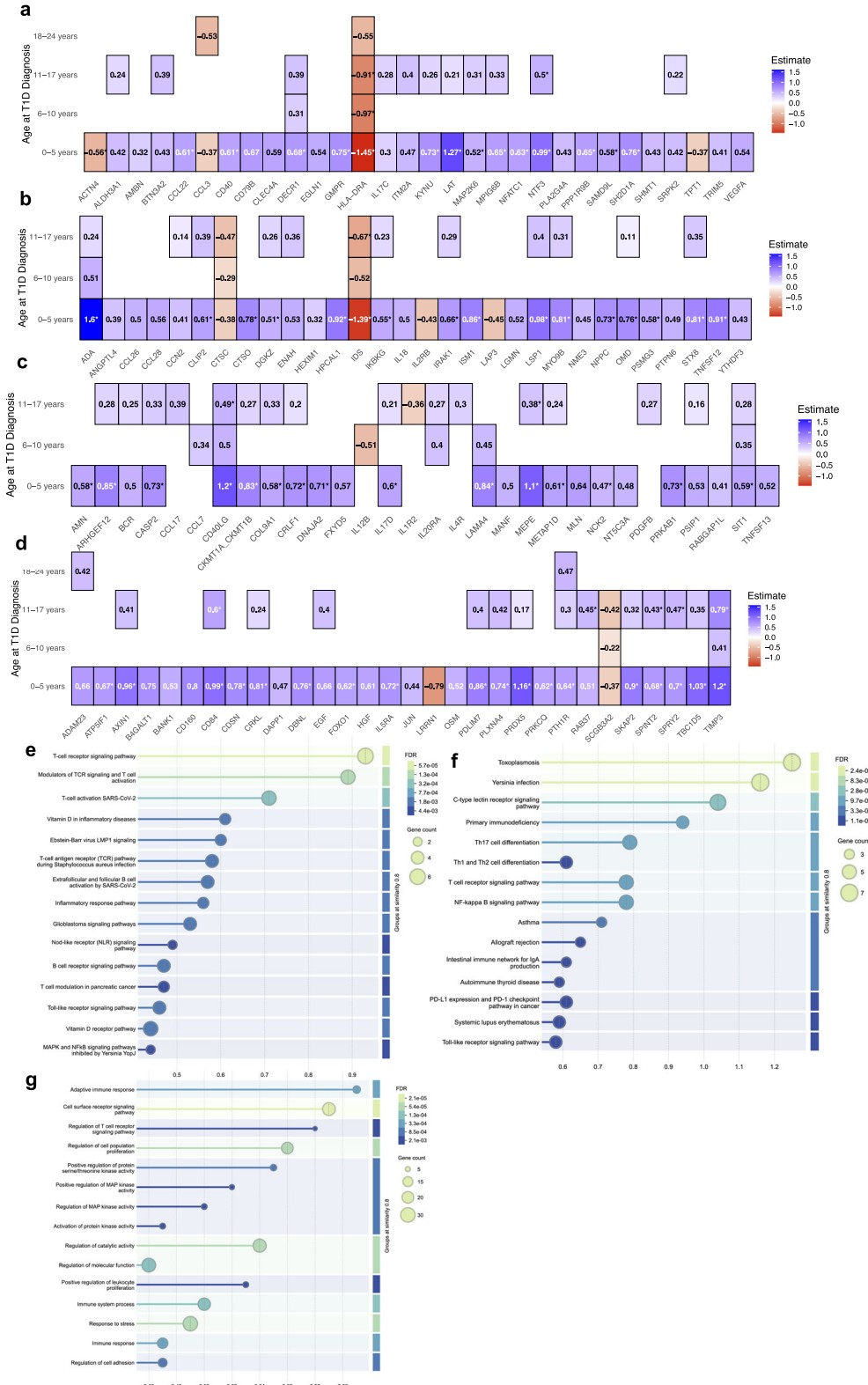

secretion through the exocytotic process. Importantly, this lysosomal enzyme is expressed in pancreatic islets and its overexpression in insulinoma-derived cell line (INS-1E) rat β-cells results in heightened insulin secretion[35]. CTSC, a lysosomal cysteine protease higher in the global T1D analysis, plays an important role in immune defenses, apoptosis, antimicrobial activity, and pro-inflammatory responses[36]. CTSC variants have been implicated in 120 human diseases[37].

That protein abundance differences occur even in the absence of HLA genetic risk suggests that the role of inflammation goes beyond genetic predisposition. Both IDS and HLA-DRA are higher in future T1D, irrespective of the presence of HLA risk alleles, while TIMP3 is reduced, except in the subset with decreased or neutral HLA-mediated T1D risk. Notably, our RFE-based prediction models perform similarly whether or not DR4DQ8 and DR3DQ2 dosage is included. High-risk

**Fig. 4 | Significant differences in normalized protein abundance by age at type 1 diabetes (T1D) diagnosis. a–d** Differences in protein abundance among controls ($n = 286$) and T1D stratified by groups based on age at T1D diagnosis: 0–5 years ($n = 23$), 6–10 years ($n = 34$; >5–10 years), 11–17 years ($n = 60$; >10–17 years), and 18–24 years ($n = 29$; >17–24 years), with $p$ values shown and asterisks indicating significance from Wilcoxon tests using the Olink Analyze R package, after false discovery rate (FDR) correction. For all significant proteins, see Supplementary Data 8e–g) Pathways significantly enriched among proteins differing most in the 0–5 year group, identified using the STRING database (similarity threshold ≥0.8). FDR-adjusted $p$-values are shown. **e** Wikipathways; **f** KEGG pathways; **g** Gene Ontology: Biological Process. Functional enrichment results are provided in Supplementary Data 9. Abbreviations for the proteins marked by asterisks in the plot ($p < 0.05$ after FDR correction): ADA adenosine deaminase, ACTN4 actinin alpha-4, AMN amnionless, ARHGEF12 Rho guanine nucleotide exchange factor 12, ATP5IF1 ATP synthase inhibitory factor 1, AXIN1 axin-1, CASP2 caspase-2, CD40 CD40 molecule, CD40LG CD40 ligand, CD84 CD84 molecule, CDSN corneodesmosin, CKMT1A_CKMT1B creatine kinase, mitochondrial 1A/1B, CLP2 caseinolytic mitochondrial matrix peptidase proteolytic subunit, CNAJA2 cochaperone Cdc37 homolog A2, COL9A1 collagen type IX alpha-1, CRKL CRK-like protein, CRLF1 cytokine receptor–like factor 1, CTSO cathepsin O, DBNL drebrin-like protein, DECR1 2,4-dienoyl-CoA reductase 1, DGKZ diacylglycerol kinase zeta, FOXO1 forkhead box protein O1, GMPR guanosine monophosphate reductase, HLA-DRA major histocompatibility complex, class II, DR alpha, HPCAL1 hippocalcin-like protein 1, IDS iduronate 2-sulfatase, IKBKG inhibitor of kappa-B kinase regulatory subunit gamma, IL5RA interleukin-5 receptor subunit alpha, IL17D interleukin-17D, IRAK1 interleukin-1 receptor–associated kinase 1, ISM1 ischemia-induced mitogen 1, KYNU kynureninase, LAMA4 laminin subunit alpha-4, LAT linker for activation of T cells, LSP1 lymphocyte-specific protein 1, MAP2K6 mitogen-activated protein kinase kinase 6, MEPE matrix extracellular phosphoglycoprotein, METAP1D methionine aminopeptidase type 1D, MPIG6B megakaryocyte and platelet inhibitory receptor G6b, MYO9B myosin-IXb, NCK2 NCK adaptor protein 2, NFATC1 nuclear factor of activated T cells 1, NPPC natriuretic peptide C, NTF3 neurotrophin-3, OMD osteomodulin, PDLIM7 PDZ and LIM domain protein 7, PLXNA4 plexin A4, PPP1R9B protein phosphatase 1 regulatory subunit 9B, PRDX5 peroxiredoxin-5, PRKAB1 AMP-activated protein kinase subunit beta-1, PRKCQ protein kinase C theta, PSMG3 proteasome assembly chaperone 3, PTH1R parathyroid hormone 1 receptor, RAB37 RAB37 GTPase, SAMD9L sterile alpha motif domain–containing protein 9-like, SH2D1A SH2 domain–containing protein 1A, SIT1 signaling threshold-regulating transmembrane adapter 1, SKAP2 src kinase-associated phosphoprotein 2, SPINT2 serine protease inhibitor, Kunitz type 2, SPRY2 sprouty RTK signaling antagonist 2, STX8 syntaxin-8, TBC1D5 TBC1 domain family member 5, TIMP3 tissue inhibitor of metalloproteinases 3, TNFSF12 TNF superfamily member 12.

HLA dosage adds little predictive value beyond what is achieved by HLA-DRA protein abundance. Associations with proteins involved deterministically with T-helper lymphocytes supports the critical hypothesis first linking β-cell and T-cytotoxic lymphocytes to environmental or autologous antigenic peptides[27]. After adjustment for strong confounds like family history of T1D, protein differences persist, with an additional marker emerging—PRSS8. This serine protease is involved in glucose-dependent regulation of insulin secretion in pancreatic β-cells through epidermal growth factor—epidermal growth factor receptor (EGF–EGFR) signaling[38] and modulates hepatic insulin sensitivity via TLR4 signaling[39].

Proteomic differences also point to potential disruptions of glycosaminoglycans (GAG). The ECM is an active participant in inflammation in local tissues, whereby a healthy ECM supports β-cell homeostasis and, conversely, inflammatory environments make the islet ECM vulnerable to infiltration and, consequently, β-cell damage and insulitis[40]. Evidence of insulitis associates with the progressive loss of this tissue is supported by both histologic findings and experimental models[41]. Components, including a common glycosaminoglycan, hyaluron/hyaluronic acid, have been investigated, as well as enzymes promoting their degradation[41,42]. GAGs provide resilience, orchestrating immunity and interacting with growth factors and cytokines, and maintain the structure, function, and hydration of various tissues, influencing tissue repair. ECM remodeling rates are particularly high in inflammatory conditions, with components fragmented and deposited in excess, resulting in a positive-feedback loop of recruitment of immune cells for proper repair and recycling[42]. The fragments act as chemoattractants for leukocytes, further implicating their role in chronic inflammation. IDS forms complexes with other molecules to bolster the ECM and is responsible for the degradation of heparan sulfate and dermatan sulfatate, both of which are GAGs critical to the ECM and proper cell signaling[43]. Enzymes involved in GAG degradation are upregulated during chronic inflammation, whereby GAG accumulation can influence lysosomal activity. Remarkably, a recent study also using Olink panels found significantly increased IDS in adult patients with T1D for ten or more years[44]. In the same study, FXYD domain-containing ion transport regulator 5 (FXYD5, also called dysadherin) is significantly reduced in T1D patients with remaining C-peptide[44]; this protein is also identified by our RFE gradient boosted model (reduced in those with future T1D). Macrophage inflammatory proteins CCL3 and CCL4, which are proinflammatory and bind to heparan sulfate, are found here to be higher in future T1D. Both are upregulated in first-degree relatives of patients with T1D across the lifespan and are also associated with the presence of multiple islet autoantibodies[45]. Notably, impaired insulin-degrading enzyme (IDE) activity results in the accumulation of these proteins[46].

The starkest contrast is observed in children diagnosed prior to age five (Fig. 4a–c). Functional pathway enrichment highlights pathways governing infection responses and NF-κB activation (an event early in T1D pathogenesis[47]). Although rare, some of the significant proteins are involved in *Yersinia* infection, which has been implicated in autoimmune thyroid disease[48] and pancreatitis[49]. The results here do not necessarily indicate infection but rather underscore the potential role of activation of similar pathways involved in B-cell mitogenic activity[50]. Blockade of cytokine-induced activation of NF-κB reduces nitric oxide free radicals and β-cell death[51]. The pathways mediating apoptosis are complex, but NF-κB activation is a key event. Higher in controls, MAP2K6 and inhibition of NF-κB kinase regulatory subunit gamma[52] (IKBKG) are common among the protein networks identified. In response to cytokines and environmental stress[53], MAP2K6 activates the p38 MAP kinase[54], which regulates the uptake of glucose induced by insulin[51] and is involved in many cellular processes, including apoptosis.

Possibly indicative of protective effects, TIMP3, ADA, and LSP1 are lower in controls. In streptozotocin diabetes-induced mice, the overexpression of TIMP3, which maintains islet architecture and functions, resulted in improved insulin secretion, vascularization, antioxidant defense, and islet morphology, as well as reduced pro-inflammatory cytokines TNF, pro-inflammatory cytokine interleukin 1 beta (IL-1β), and cytokine interferon gamma (IFN-γ[55]). In human tissue cultures, TIMP3 blocks the release of TNF, preventing spontaneous inflammation[56]. ADA deaminates adenosine, an immunosuppressive signal that prevents excess inflammatory response[57]. Its secretion by macrophages and lymphocytes enhances autoantibody production[58]. LSP1 facilitates transendothelial neutrophil migration[59] and may help prevent neutrophil accumulation in the pancreas[60]. Circulating neutrophil counts correlate with β-cell destruction in T1D, where NETosis, the release of neutrophil extracellular traps (NET), has also been shown to damage tissue in a manner that is not yet understood[61,62].

The SHAP-based protein-metabolite/exogeneous compound analysis reveal non-linear relationships between metabolites, exogenous comounds, and protein abundance in cord blood. In controls, elevated protein expression is associated with metabolites and compounds involved in amino acid metabolism (lysine, proline), bile acid

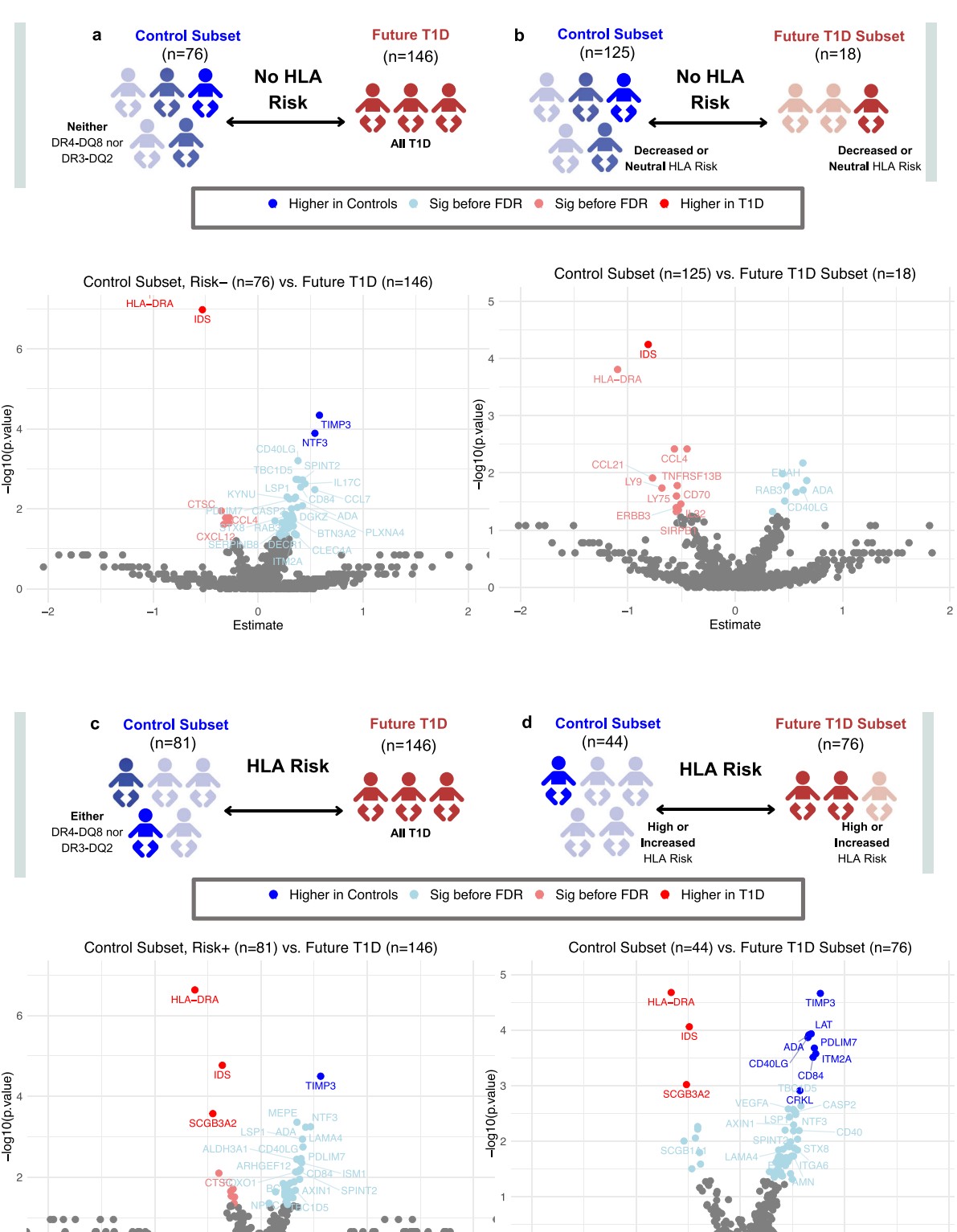

derivatives (tauro-β-muricholic acid, 7-oxo-DCA), and lipid metabolism (palmitoleic acid, arachidic acid). Notably, ADA shows a strong positive association with itaconic acid, an immunomodulator[63] that has antimicrobial impacts and regulates macrophages and reactive oxygen species[64]. ADA is inversely associated with arachidonic acid, a polyunsaturated fatty acid that is released after tissue injury and leads to the formation of eicosanoids, contributing to inflammation and

peripheral nervous system excitation[65], with both positive and negative effects on β-cell function depending on the context[66]. Elevated arachidonic acid has been associated with future T1D autoimmunity in a Finnish birth cohort[67] as well as bipolar disorder in postmortem studies, alongside stearic acid levels[68].

Conversely, proteins elevated in future T1D show strong positive associations with environmental contaminants (like perfluorinated

**Fig. 5 | Proteomic differences in infants with future type 1 diabetes (T1D) and controls, stratified by HLA genetic risk for T1D.** Proteomic differences between future cases and controls, stratified by HLA genetic risk types for T1D, based on Wilcoxon tests using the Olink Analyze R package. Non-HLA genetic risk comparisons: **a** Significant differences in protein abundance comparing controls without DR4-DQ8 or DR3-DQ2 alleles (*n* = 76), to all future T1D cases (*n* = 146); **b** Controls with decreased or neutral risk (*n* = 125) compared to future T1D with decreased or neutral risk (*n* = 18). HLA genetic risk comparisons; **c** Controls possessing either DR4-DQ8 or DR3-DQ2 (*n* = 81) compared to all future T1D cases (*n* = 146); **d** Controls with increased or high-risk (*n* = 44) compared to future T1D with increased or high-risk (*n* = 76). Normalized protein expression (NPX) levels are shown, with significance assessed using Wilcoxon statistics in the OlinkAnalyze R package. *P* values before and after false discovery correction (FDR) are indicated. ADA adenosine deaminase, ALDH3A1 aldehyde dehydrogenase 3 family member A1, AMN amnionless, ARHGEF12 Rho guanine nucleotide exchange factor 12, AXIN1 axin-1, BTN3A2 butyrophilin subfamily 3 member A2, CASP2 caspase-2, CCL4 C-C motif chemokine ligand 4, CCL7 C-C motif chemokine ligand 7, CCL21 C-C motif chemokine ligand 21, CD40LG CD40 ligand, CD70 CD70 molecule, CD84 CD84

molecule, CLEC4A C-type lectin domain family 4 member A, CRKL CRK-like protein, CTSC cathepsin C, CXCL12 C-X-C motif chemokine ligand 12, DECR1 2,4-dienoyl-CoA reductase 1, DGKZ diacylglycerol kinase zeta, ENAH enabled homolog, ERBB3 erb-b2 receptor tyrosine kinase 3, FOXO1 forkhead box protein O1, HLA-DRA major histocompatibility complex, class II, DR alpha, IL17C interleukin-17C, IL32 interleukin-32, ISM1 ischemia-induced mitogen 1, ITGA6 integrin subunit alpha-6, ITM2A integral membrane protein 2A, KYNU kynureninase, LAMA4 laminin subunit alpha-4, LSP1 lymphocyte-specific protein 1, LY9 lymphocyte antigen 9, LY75 lymphocyte antigen 75, MEPE matrix extracellular phosphoglycoprotein, NTF3 neurotrophin-3, PDLIM7 PDZ and LIM domain protein 7, PLXNA4 plexin A4, RAB37 RAB37 GTPase, SCGB1A1 secretoglobin family 1A member 1, SCGB3A2 secretoglobin family 3A member 2, SERPINB8 serpin family B member 8, SIRPB1 signal-regulatory protein beta-1, SPINT2 serine protease inhibitor, Kunitz type 2, STX8 syntaxin-8, TBC1D5 TBC1 domain family member 5, TIMP3 tissue inhibitor of metalloproteinases 3, TNFRSF13B tumor necrosis factor receptor superfamily member 13B, VEGFA vascular endothelial growth factor A. Created in BioRender. Ahrens, A. (2025) https://BioRender.com/k05i837.

environmental contaminant perfluorooctanesulfonate, linear, PFOS-L) and saturated fatty acids (stearic acid), alongside varying influences from bile acid metabolites and amino acids (e.g., glutamine). The associations with PFOS-L are of particular interest, where we see trends between PFOS-L and immune-related proteins, including IL18, TNFRSF4, and IL17A. Because they point to pathways involved in inflammation, cytotoxicity, and immune regulation, these directional associations suggest that even environmentally relevant PFOS exposures may subtly modulate immune signaling[69], consistent with known immunomodulatory effects of PFOS, warranting further investigation into potential impacts on health. These findings suggest that even low-level environmental exposure may subtly influence immune signaling as early as birth. Lower levels of stearic acid are seen in infants with the highest expression of ADA and TIMP3. Stearic acid has been shown to induce CD11c+ macrophage differentiation and to promote inflammation via increased expression of the costimulatory molecules cluster of differentiation 80 (CD80) and CD86 and of cytokines IL-6, TNF, and IL-1β, in an epidermal fatty acid binding protein (E-FABP)-dependent manner[70]. These fatty acid binding proteins are known to be higher in T1D patients[71]. They are correlated with islet autoantibody titer[72] and are predictive of the progression of diabetic neuropathy irrespective of stage of T1D[73]. Elevated stearic acid is found in one year olds with future islet autoimmunity in the Trial to Reduce IDDM in the Genetically at Risk (TRIGR) study, even after adjustment for HLA and maternal T1D[74]. Environmental exposures, when combined with proteomic changes, may contribute to disease development. In our study, perfluorinated environmental contaminant compounds are positively correlated with HLA-DRA and IDS, both significantly higher at birth in infants with future T1D, regardless of genetic risk. Prior research has shown that prenatal perfluorinated environmental contaminant exposure can modulate phospholipids and bile acids and is linked to increased risk of islet autoimmunity, especially in those with T1D-associated HLA genotype, likely resulting from direct fetal exposure to PFOS[75].

Supervised ML prediction models perform well based on these protein markers at birth, with the highest performance of AUC = 0.89. Gradient boosting has been applied in several ML areas, superior for its ability to avoid overfitting and deal with complexity in disease prediction[76]. Our results demonstrate robust prediction using proteomics. Interestingly, the preponderance of the performance of these models is not dependent on HLA genotype, indicating that the predictive value of the proteins extends beyond HLA. This finding is particularly important given ongoing debates about genetic screening at birth.

Expanding the dataset and testing this model in a large independent cohort will provide evidence of generalizability. While the ML

models performed well, further optimization is possible through exploration of other non-HLA genetic and environmental factors, that may influence inflammation and T1D risk. Also, the proteomic analysis is performed here does not cover the entire proteome, leaving the possibility of key proteins or pathways being unaccounted for. While efforts are made to account for potential confounders, the influence of unmeasured factors cannot be ruled out. As the ABIS study is limited to children born in Sweden between 1997 and 1999—a population that may not represent broader demographic and environmental variability, the generalizability in other populations must be investigated. Nonetheless, this study provides evidence to facilitate continued research into the impacts of early inflammation during early childhood.

Once childhood islet autoantibodies are detected, the inflammatory attack destroying pancreatic islet cells is already in progress, making effective interventions at this late stage more difficult. Intervening prior to the appearance of islet autoantibodies is critical. Our findings highlight the predictive potential of birth-associated proteins involved in cell death, immune response, cytoskeletal dynamics, and response to abiotic stress—suggesting that the prenatal inflammatory environment may influence disease trajectory. Biomarkers centered around the prenatal period could be transformative, offering the opportunity to reduce inflammation and more potentially delay or prevent the onset of insulin dependence.

## Methods
### Participants
The ABIS cohort follows, to present day, 16,683 children born between October 1997 and 1999 in Southeastern Sweden. Families are recruited from nine obstetric clinics across all hospitals in the counties of Östergötland, Småland, Blekinge, and Öland. They receive oral and written information and are invited to join the study. Out of the 21,700 children born in the region, 78.6% of families provided informed consent for their child to participate. Cord blood samples were collected from all ABIS children. Parents completed extensive questionnaires and diaries from the child's birth through 13 years of age, with ABIS individuals themselves answering questionnaires in addition to parents at 8, 11–13, and then themselves at 17–19 and 23–26 years of ages[77]. Only the birth questionnaire is analyzed in the present study, focused on pregnancy and birth.

Biological sex (male/female) is collected in the ABIS study from the birth questionnaire (reported by the parent) and is included in the study design. Gender identity is not collected; therefore, all analyses pertain solely to biological sex. The ABIS cohort shows a relatively balanced distribution of biological sex (48.2% female, 51.8% male), and the distribution of biological sex in the T1D case and control groups is

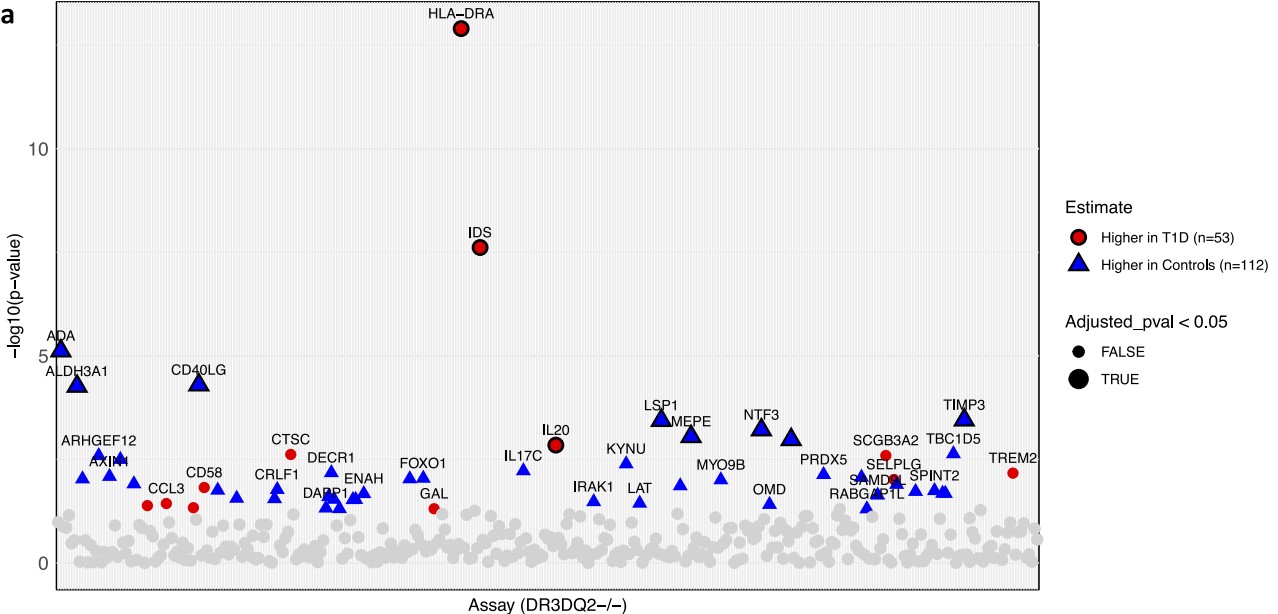

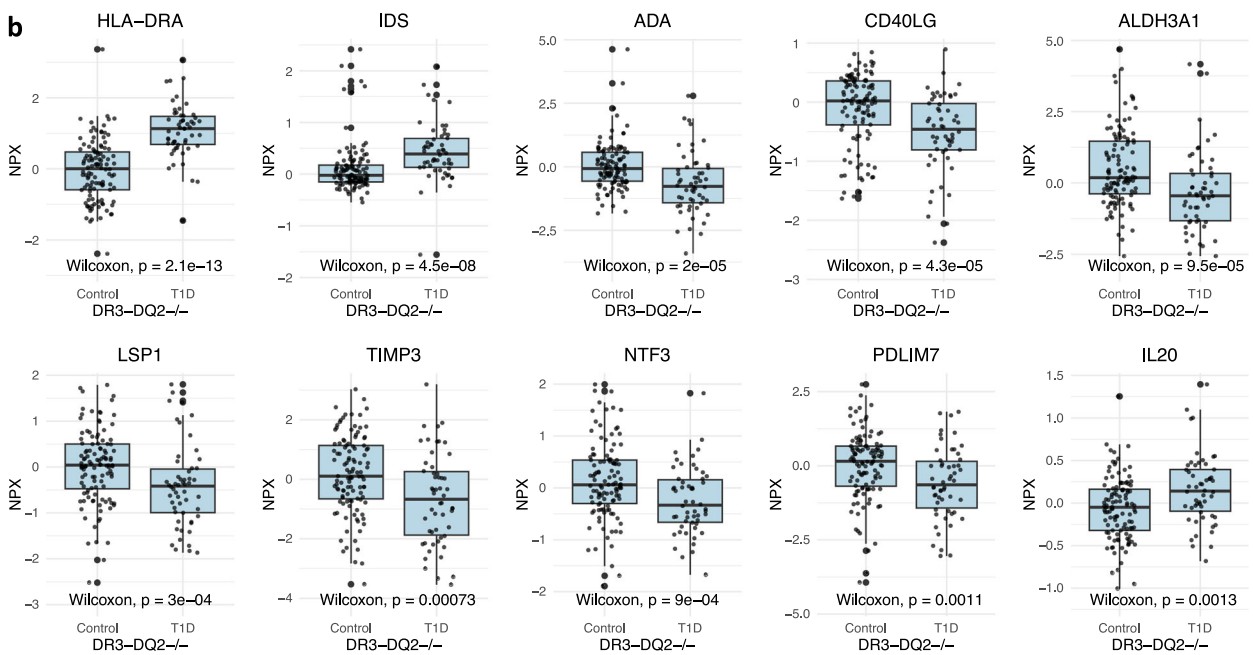

**Fig. 6 | Proteomic differences in type 1 diabetes (T1D) and controls, specifically those lacking DR3-DQ2 HLA alleles. a** Significant proteomic markers in infants with future type 1 diabetes (T1D; *n* = 53) and controls (*n* = 112), all of whom lack the DR3-DQ2 human leukocyte antigen (HLA) allele. Proteins that remain significant after false discovery rate (FDR) correction are indicated with bold outlines; **b** Boxplots showing the most significant proteins, with Wilcoxon *p* values with a two-sided test indicated. Boxplots show the median (line) and interquartile range (box, 25th–75th percentile) to demonstrate the data distribution. Whiskers extend to the most extreme values within 1.5 × interquartile range (IQR) of the lower and upper quartiles; points beyond the whiskers are plotted as outliers. Values are displayed as normalized protein expression (NPX) values. ADA adenosine deaminase, ALDH3A1 aldehyde dehydrogenase 3 family member A1, ARHGEF12 Rho guanine nucleotide exchange factor 12, AXIN4 axin family member 4, CCL3 C-C

motif chemokine ligand 3, CD40LG CD40 ligand, CD58 CD58 molecule, CRLF1 cytokine receptor-like factor 1, CTSC cathepsin C, DAPP1 dual adaptor of phos-photyrosine and 3-phosphoinositides 1, DECR1 2,4-dienoyl-CoA reductase 1, ENAH enabled homolog, FOXO1 forkhead box protein O1, GAL galanin peptide, IDS iduronate 2-sulfatase, IL17C interleukin-17C, IL20 interleukin-20, IRAK1 interleukin-1 receptor-associated kinase 1, KYNU kynureninase, LAT linker for activation of T-cells, LSP1 lymphocyte-specific protein 1, MEPE matrix extracellular phosphogly-coprotein, MYO9B myosin-IXB, NTF3 neurotrophin-3, OMD osteomodulin, PRDX5 peroxiredoxin-5, RABGAP1L RAB GTPase-activating protein 1-like, SAMD9L sterile alpha motif domain-containing protein 9-like, SCGB3A2 secretoglobin family 3A member 2, SELPLG selectin P ligand, SPINT2 serine protease inhibitor, Kunitz type 2, TBC1D5 TBC1 domain family member 5, TIMP3 tissue inhibitor of metallopro-teinases 3, TREM2 triggering receptor expressed on myeloid cells 2.

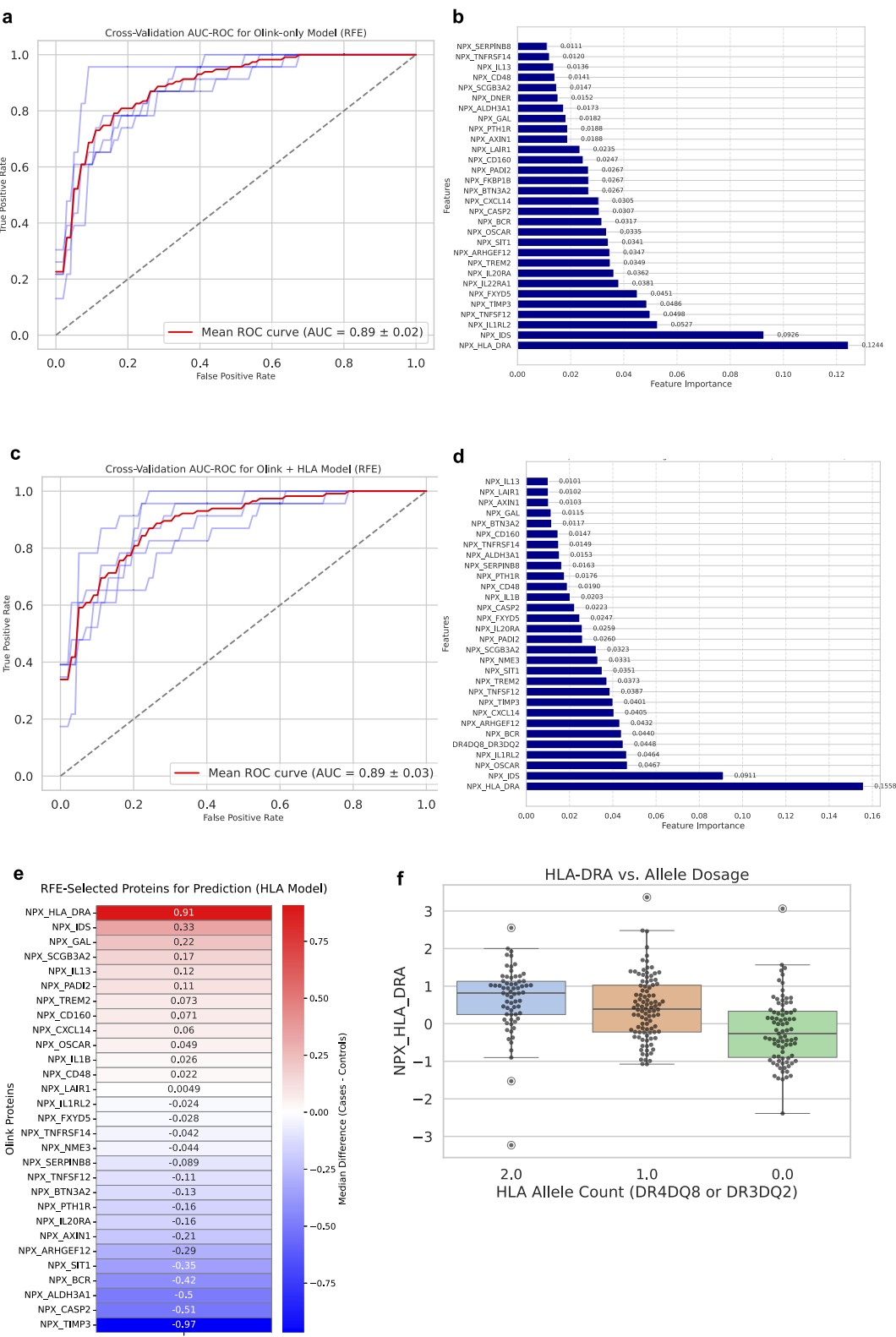

broadly representative, but with a slight overrepresentation of females among controls included in the Olink proteomics subset (52.5%) compared with controls not included (47.9%). No socially constructed or socially relevant variables, such as race, ethnicity, or socioeconomic status, are collected in this study. The ABIS cohort consists primarily of children born in Sweden to Swedish parents, and although data on race, ethnicity, or ancestry are not gathered, the population is

expected to be relatively homogeneous. At enrollment, 89.2% of ABIS children have both parents born in Sweden, 7.7% have one parent born outside Sweden, and 3.1% have both parents born outside Sweden[78]. These data come from self-report, and serve only to contextualize the demographic structure of the cohort rather than classify participants into sociocultural groups. Because these variables are not collected,

**Fig. 7 | Supervised machine learning (ML) models to predict type 1 diabetes (T1D) from recursive feature elimination (RFE)-selected proteins, with and without inclusion of high-risk HLA allele dosage. a** Performance of the Olink-only model (without HLA dosage), using eXtreme Gradient Boosting (XGBoost), based on **b** the top 30 recursive feature elimination (RFE)-selected proteins using an 80% training and 20% testing set from in the ABIS Olink case/control cohort ($n = 432$; cases, $n = 146$; controls, $n = 286$). **c** Performance of the Olink + HLA machine learning (ML) model, based on **d** the top 30 RFE-selected features. **e** Median difference of the RFE-selected proteins in the Olink + HLA model. **f** Normalized protein expression (NPX) values of HLA-DRA by high-risk HLA allele dosage (0, 1, or 2 copies of DR4DQ8 or DR3DQ2). Boxplots show the median (line) and interquartile range (box, 25th–75th percentile) to demonstrate the data distribution. Whiskers extend to the most extreme values within 1.5 × interquartile range (IQR) of the lower and upper quartiles; points beyond the whiskers are plotted as outliers. AUC-ROC area under the receiver operating characteristic curve, ALDH3A1 aldehyde dehydrogenase 3 family member A1, ARHGEF12 Rho guanine nucleotide exchange factor

12, AXIN1 axis inhibition protein 1, BCR breakpoint cluster region protein, BTN3A2 butyrophilin subfamily 3 member A2, CASP2 caspase-2, CD48 CD48 molecule, CD160 CD160 molecule, CXCL14 C-X-C motif chemokine ligand 14, DNER delta and notch-like epidermal growth factor-related receptor, FXYD5 FXYD domain-containing ion transport regulator 5, GAL galanin and GMAP prepropeptide, HLA_DRA major histocompatibility complex, class II, DR alpha, IDS iduronate 2-sulfatase, IL1B interleukin-1 beta, IL13 interleukin-13, IL20RA interleukin-20 receptor subunit alpha, IL22RA1 interleukin-22 receptor subunit alpha-1, IL1RL2 interleukin-1 receptor-like 2, LAIR1 leukocyte-associated immunoglobulin-like receptor 1, NME3 NME/NM23 family member 3, OSCAR osteoclast-associated receptor, PADI1 peptidyl arginine deiminase 1, PADI2 peptidyl arginine deiminase 2, PTH1R parathyroid hormone 1 receptor, SCGB3A2 secretoglobin family 3A member 2, SERPINB8 serpin family B member 8, SIT1 signaling threshold-regulating transmembrane adapter 1, TIMP3 tissue inhibitor of metalloproteinases 3, TNFRSF14 tumor necrosis factor receptor superfamily member 14, TNFSF12 tumor necrosis factor ligand superfamily member 12, TREM2 triggering receptor expressed on myeloid cells 2.

we do not use race or ethnicity as proxies for any other social variable, and no analyses rely on such constructs.

### Diagnoses and study design
Diagnoses are obtained using International Classification of Diseases (ICD) codes as reported in the Swedish National Patient Register, which includes records from specialist consultations in outpatient care from public and private providers. Up to December 2020, a total of 167 ABIS children had received a T1D diagnosis (ICD-10 code E10), with diagnoses occurring at an average of 12.6 ± 6.1 years, ranging from 2 to 24.6 years (median: 12.6 years) and a cumulative incidence of 1%. Controls are defined as individuals without any autoimmune or neurodevelopmental diagnoses in their medical record, or according to the Swedish National Patient Register.

This study uses observational human cohort data with no experimental intervention. Group allocation (future T1D vs. control) was determined independently through clinical follow-up using predefined diagnostic criteria and not assigned by investigators. Blinding was not applicable, as predefined clinical outcomes and standardized laboratory measurements were used. Group status (future T1D vs. control) was determined independently through clinical follow-up, and all analyses were conducted using objective computational methods without subjective assessment. No formal sample size calculation was performed. This study leveraged all cord blood samples from children who later developed T1D in the ABIS cohort available at the time of analysis, along with eligible controls with sufficient material. As this was an observational cohort study using existing biobank material, sample sizes were determined by sample availability and are consistent with prior multi-omic studies of early-life T1D risk.

### Birth questionnaires
The psychosocial vulnerability index, as used in Ahrens et al.[78], is derived from questions pertaining to the following: living conditions, parental education levels, employment and income, stressful life events, social support, and safety (Table 2).

### Olink inflammatory and immune panels
Olink's Explore 384 Inflammation 1 and 2 panels and Target Immune Response panel are used to analyze cord serum samples ($n = 1204$) at Olink's headquarters in Uppsala, Sweden, encompassing 286 controls, 147 with future T1D, as well as samples from children with other conditions in the future. Controls are identified as individuals without any future diagnoses of autoimmune diseases, psychiatric disorders, or neurodevelopmental conditions, whereas cases consisted of individuals with confirmed T1D diagnoses by 22 years of age. Control subjects are randomly selected from the broader ABIS study population. To address the notable class imbalance, oversampling of the minority

class and undersampling of the majority class are performed. Future T1D case and control groups are broadly representative of the full ABIS cohort, with no major evidence of selection bias (Table 1). The only notable difference is a modest overrepresentation of females among controls included in the Olink subset (52.5%) versus those controls who are not (47.9%).

The Olink Explore panels include 370 inflammatory protein markers, with an additional 14 proteins for quality control. These markers encompass a range of chemokines and cytokines, such as interferon gamma, IL-1α, IL-6, IL-10, IL-12, IL-13, IL-15, IL-17, and LTa. The Olink Target Immune Response panel comprises 92 protein biomarkers, focusing on adaptive immune responses, viral responses, and T-cell proliferation. The following assays do not meet Olink's batch release quality control criteria and are therefore not included in this project: BCL2L11, BID, LTA, GZMB, MGLL, FLI1, MPI, EBI3-IL27, and ANGPTL7.

### Quality control in proteomic analysis
Internal and external controls are run to ensure that the data does not suffer from batch effects of other technical variations. Three internal controls are added to each sample: the Incubation control, the Extension Control, and the Amplification control. The Extension Control is used for the generation of the NPX values. The Incubation Control and the Amplification Control are used to monitor the quality of assay performance, as well as the quality of individual samples. In addition, three external controls are included in each run, the Plate Control (healthy pooled plasma), Sample Control (healthy pooled plasma), and Negative Control. The Plate Control is used for data normalization, the Sample Control is used to assess potential variation between runs and plates, and the Negative Control is used to calculate Limit of Detection (LOD) for each assay and to assess potential contamination of assays. To pass QC, there should be at least 500 average matched counts (reads for each specific combination of sample and assay). Also, the deviation of the median of the Negative Controls must be less or equal to 5 standard deviations from the set predefined value for that assay, otherwise the assay receives a warning status. Two of the 1204 samples do not meet quality control, resulting in a final dataset of 1202 ABIS individuals.

### Pre-processing of proteomic data
The proteomic data is presented as normalized protein expression (NPX) values. The NPX is Olink's relative protein quantification unit on log2 scale. These values are calculated from the number of matched counts, using Next Generation Sequencing as readout. Data values for measurements below LOD are reported for all samples. All data are intensity-normalized, with the exception of PNLIPRP2 and FOLR3, which are plate control normalized because these assays showed a

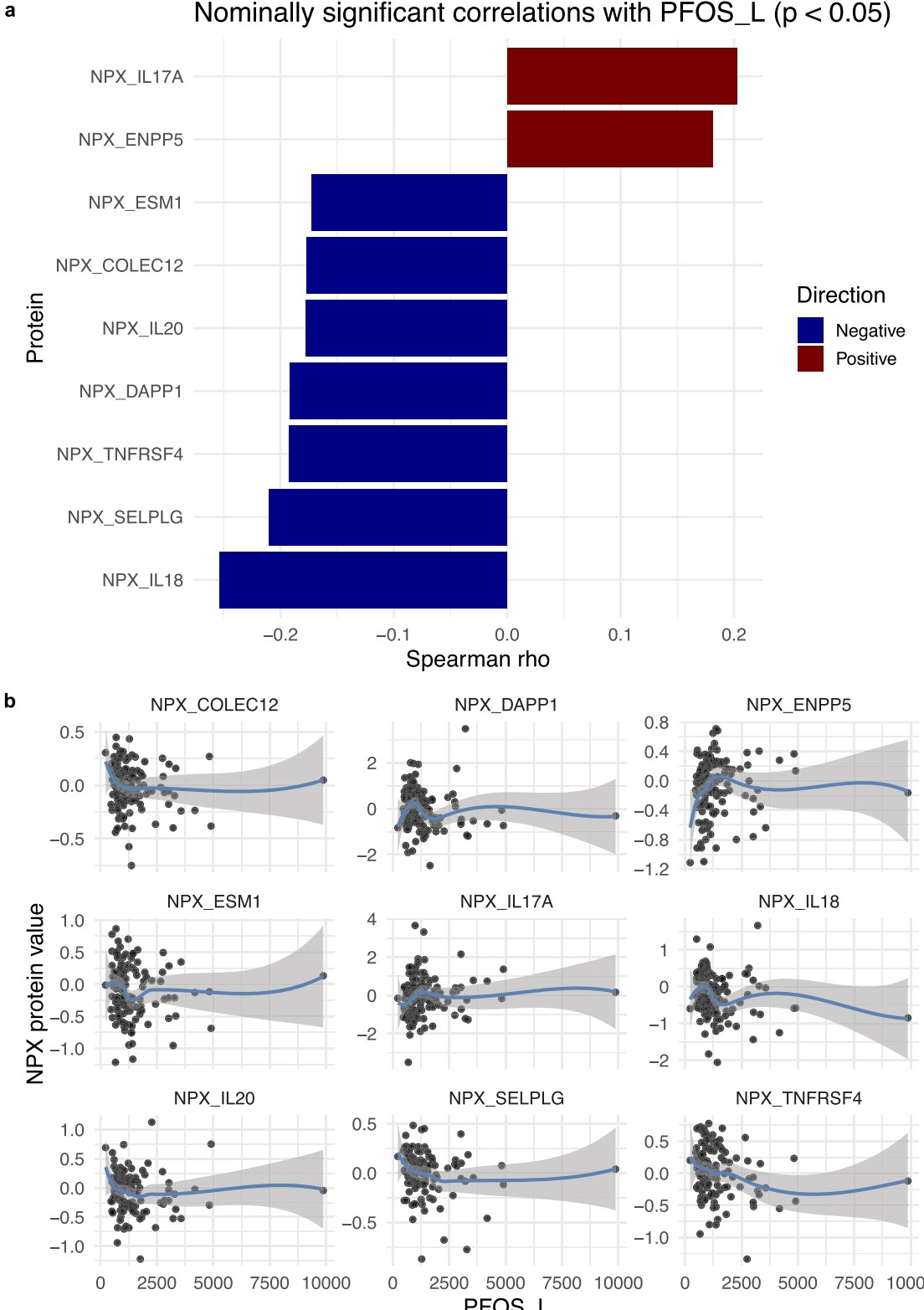

**Fig. 8 | Spearman correlations between circulating proteins and PFOS-L in plasma (*n* = 132). a** Spearman rank correlations between PFOS-L, the linear isomer of perfluorooctane sulfonate, and circulating immune and metabolic proteins (NPX values). Only proteins with nominal *p* < 0.05 are shown, with negative correlations in blue and positive correlations in red. **b** Scatterplots illustrating the relationship between PFOS-L and each protein for individual participants, highlighting directional trends. All correlations were assessed using a two-sided test.

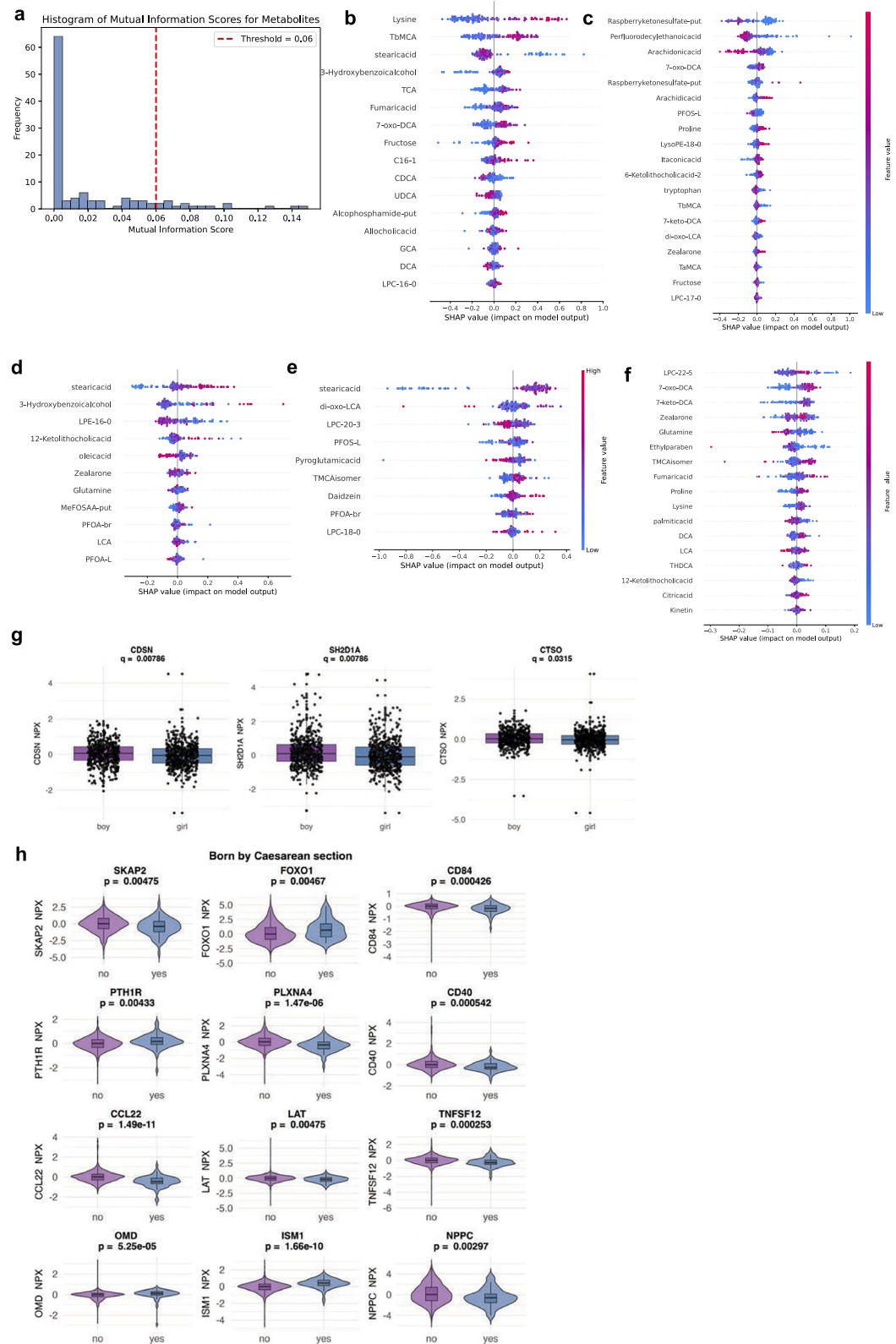

natural bimodal distribution. In >50% of the samples, 96% of the proteins from the Explore 384 Inflammation are detected.

### Metabolomic data

Cord serum samples are analyzed by lipidomics and hydrophilic (water-soluble) metabolite profiling[78]. For lipidomic analysis, 10 μL of serum is mixed with internal standards and analyzed by ultra-high-

performance liquid chromatography quadrupole time-of-flight mass spectrometry (UHPLC-QTOFMS from Agilent Technologies; Santa Clara, CA, USA). For polar metabolites and exogenous compounds, 40 μL of serum is mixed with internal standards and analyzed by the Agilent 1290 Infinity LC system coupled with 6545 Q-TOF MS interfaced with a dual jet stream electrospray (dual ESI) ion source (Agilent Technologies, Santa Clara, CA, USA). Quality controls include blanks,

**Fig. 9 | Associations between the cord blood metabolome, exogenous compounds, pre and perinatal factors, and key type 1 diabetes (T1D)-associated proteins.** Metabolites measured in cord blood at birth ($n$ = 132) are screened for possible association with proteins, using mutual information (MI) scoring. The elbow method sets the MI threshold for selecting significant metabolites. **a** For tissue inhibitor of metalloproteinases 3 (TIMP3), the top 16 metabolites are shown. **b**–**f** SHAP (SHapley Additive exPlanations) values illustrate relationships between these metabolites and the concentration levels of key T1D-associated proteins in cord blood ($n$ = 132). Metabolites linked to **b** TIMP3 and **c** adenosine deaminase (ADA), which are both decreased in children who later develop T1D. Metabolites linked to **d** iduronate 2-sulfatase (IDS), **e** major histocompatibility complex, class II, DR alpha (HLA-DRA), and **f** cathepsin C (CTSC), which are elevated in children who later develop T1D. Protein abundances (normalized protein expression, NPX, values) are shown, based on Wilcoxon and eXtreme Gradient Boosting (XGBoost) analyses. Factors associated by two-sided test with concentrations of proteins identified in Fig. 3c, d and Supplementary 4, including **g**) biological sex, **h** birth by Caesarean section, all significant after FDR correction for multiple comparisons ($n$ = 1204). The boxplot (**g**) shows the median (line) and interquartile range (box, 25th–75th percentile) to demonstrate the data distribution. Whiskers extend to the most extreme values within 1.5 × interquartile range (IQR) of the lower and upper quartiles; points beyond the whiskers are plotted as outliers. Values are displayed as normalized protein expression (NPX). 7-keto-DCA 7-ketodeoxycholic acid, 7-oxo-DCA 7-oxodeoxycholic acid, C16-1 hexadecenoic acid, CCL22 C-C motif chemokine 22, CD40 cluster of differentiation 40, CD84 cluster of differentiation 84, CDSN corneodesmosin, CTSO cathepsin O; DCA, deoxycholic acid; di-oxo-LCA, 3,12-dioxo-lithocholic acid; FOXO1, forkhead box protein O1, GCA glycocholic acid, ISM1 immunoglobulin superfamily member 1, LAT linker for activation of T cells, LCA lithocholic acid, LPC-16:0 lysophosphatidylcholine 16:0, LPC-17:0 lysophosphatidylcholine 17:0, LPC-20:3 lysophosphatidylcholine 20:3, LPC-22:5 lysophosphatidylcholine 22:5, LPE-16:0 lysophosphatidylethanolamine 16:0, MeFOSAA-put methyl perfluorooctanesulfonamido acetic acid, putative, NPPC natriuretic peptide C, OMD osteomodulin, PFOS-L perfluorooctanesulfonate, linear, PFOA-br perfluorooctanoate, branched, PFOA-L perfluorooctanoate, linear, PLXNA4 plexin A4, PTH1R parathyroid hormone 1 receptor, SH2D1A SH2 domain–containing protein 1A, SKAP2 src kinase-associated phosphoprotein 2, TaMCA tauro-α-muricholic acid, TbMCA tauro-β-muricholic acid, TCA taurocholic acid, THDCA taurohyodeoxycholic acid, TMCA isomer tauromuricholic acid isomer, UDCA ursodeoxycholic acid.

**Table 2 | Psychosocial Vulnerability Index in ABIS**

| Variable | Coding |
|---|---|
| Living conditions | 1 = flat in an apartment building |
| Father only elementary school | 1 = Yes |
| Mother only elementary school | 1 = Yes |
| Father: Unemployed | 1 = Yes |
| Mother: Unemployed | 1 = Yes |
| Both parents born abroad | 1 = Yes |
| Mother single parent | 1 = Yes |
| Serious stressful life event | 1 = Yes |
| Mother received no support during pregnancy | 1 = Yes |
| Mother not feeling safe during pregnancy | 1 = Yes |
| Much worry for chronic illness in child | 1 = Yes |
| Disposable household income yr 2000 | 1 = 10th lowest percentile |
| Disposable household income yr 2000 | Swedish krona |
| Vulnerability index7 (sum of variables) | 0–6 (6 most vulnerable) |
| Vulnerability index4 (sum of variables) | 0–3 (3 most vulnerable = 95th percentile) |

The psychosocial vulnerability index is based on 13 variables, with coding listed below. Scores of 3 on the vulnerability index 4 or 6 on the vulnerability index 7 indicate the highest vulnerability (95th percentile) in the ABIS cohort.

pure standards, extracted standards, and control plasma. Quantification is performed[78], using 7-point internal calibration curves with a custom database based on the Metabolomics Standards Initiative and lipid-class specific authentic standards. LIPID MAPS[79], a comprehensive classification framework for lipids developed by the International Lipid Classification and Nomenclature Committee, is used to organize lipids into eight classes. Categorization of triglycerides is based on the fatty acid composition, distinguishing between saturated, monounsaturated, and polyunsaturated species, and in cases where detailed structure has not been determined, carbon and double bond sums are used for the naming. Data are processed using MZmine 2.53[80].

**Association of prenatal risk factors with future T1D**
Birth questionnaires are assessed by chi square for association with future T1D diagnosis, comparing 15,732 controls to 167 cases in Python. Variables missing in 15% or more of the cases are excluded. The most significant factors are carried into ML models for T1D diagnosis prediction, training on 70% of the dataset and holding out 30% for testing,

with a random_state = 42 for reproducibility. ABIS individuals with missing data at any single feature from the list of significant features are dropped, resulting in a final dataset of $n$ = 11,648 for the ML models.

We apply Stratified K-Fold Cross-Validation with 5 splits to evaluate the performance of Logistic Regression and three ML models: eXtreme Gradient Boosting (XGBoost), Random Forest, and SVM. Class imbalance is addressed by assigning class weights (96:1) for the minority class across all models. The top 18 features identified by MI scoring are used as predictors. MI quantifies the strength of the association between two variables, measuring how much the value of one reduces uncertainty about the other, without assuming linearity, which traditional correlation-based methods may miss[81]. As MI is model-agnostic[82], it can be used as a dimension-reduction filter method before any modeling type. For each fold, the models are trained using the training set and tested on the validation set. Key performance metrics, including Accuracy, Precision, Recall, F1 score, and AUC-ROC, are calculated. Receiver Operating Characteristic (ROC) curves are generated by interpolating True Positive Rates at fixed False Positive Rate (FPR) intervals. The mean and standard deviation of each metric are computed across folds to evaluate model stability. ROC curves for each model are plotted, with shaded areas indicating ±1 standard deviation. Mean AUC-ROC scores are reported.

**Global analysis of proteomic markers, by future T1D diagnosis**
Primary comparative case/control analysis is carried out using the OlinkAnalyze R package[83]. Differentially expressed proteins are first identified using Welch 2-sample Mann–Whitney $U$ tests on normalized Olink data (NPX) at confidence 0.95 for every protein, with false discovery rate (FDR) correction for multiple comparisons by the Benjamini–Hochberg method. Gene Set Enrichment Analysis (GSEA) is performed using OlinkAnalyze on the basis of these proteins.

To adjust for significant confounds of T1D risk, median fold-change analysis is performed, but on a subset of controls, matching cases and controls in a 1:1 ratio using propensity score matching by nearest neighbor with the matchIt[84] R package. This results in a dataset of 140 future T1D and 140 matched controls, balanced on the following factors: family history of T1D (in the mother), mode of delivery, gender, week of delivery, vulnerability index, serious life event, and stomach flu during pregnancy.

**Protein concentration differences by age at T1D diagnosis**
Future cases are categorized into four age groups: diagnosis at ≤5 years ($n$ = 23), 6–10 years ($n$ = 34), 11–17 years ($n$ = 60), and 18–24 years ($n$ = 29), and each group is compared to the full set of controls. Groupings correspond to >5 up to 10 years (label 6–10 years), >10 up to

17 years (label 11–17 years), and >17 years (label 18–24 years). Separately for each of the four analyses, a non-parametric Wilcoxon rank-sum test is performed to compare NPX values between T1D cases and controls. Median NPX values are also calculated for both groups within each assay. To provide an estimate of the magnitude of group differences, Cohen's d is calculated for each assay. The sign of Cohen's d is adjusted to indicate the direction of the difference (negative if the mean NPX is higher in controls). The p-values from the Wilcoxon tests are adjusted for multiple comparisons using the Benjamini–Hochberg (BH) method to control the FDR.

Functional pathway enrichment analysis is performed with the STRING database[25] (version 12.0), focused on the significant proteins identified in the cases diagnosed by age five. The STRING database curates PPIs, both physical and functional, and spans numerous critically assessed data sources, including Biocarta, BioCyc, GO, KEGG, and Reactome databases. The scores are calculated by the probability of two proteins interacting and corrected for the probability of a random observation of occurrence[85].

### Protein concentration differences, stratified by HLA genetic risk for T1D

Future T1D cases (n = 146) are compared to two subsets of controls (81 controls with genetic risk for T1D and 76 controls without genetic risk). HLA risk for T1D is defined by the presence MHC Class II HLA risk alleles, including DR4-DQ8 (DRB1-DQA103:01-DQB103:02) and/or DR3-DQ2 (HLA-DRB103:01-DQA105:01-DQB1*02:01)[23,26], seen in as many as 90% of T1D patients[27,28], as these represent the primary risk factor for islet autoimmunity[29,30]. Next, future T1D cases are separated by HLA risk (decreased, neutral, increased, and high), based on the presence of DR4-DQ8 and DR3-DQ2[26], and compared to controls with similar genetics. Decreased or neutral HLA risk genotypes are seen in 125 controls and 18 with future T1D, while high or increased risk identified in 44 controls and 76 with future T1D. For each of these analyses, NPX values are compared by 2-sample Mann–Whitney $U$ at confidence 0.95 for every protein, again with FDR correction.

Protein differences unique to HLA subtypes are then explored, first by individuals lacking DR3-DQ2 (n = 53 T1D, n = 112 controls), and then by individuals lacking DR4-DQ8. This analysis is focused around a specific HLA genetic background (e.g., excluding individuals with certain genotypes). For the DR3-DQ2 −/− analysis, any case or control possessing DR3-DQ2 is removed from the dataset (n = 53 cases, n = 112 controls), and for the DR4-DQ8 −/– analysis, any case or control possessing DR4-DQ8 is removed (n = 24 cases, n = 111 controls). Each filtered dataset is passed through to a custom Wilcoxon rank-sum test in the OlinkAnalyze R package, olink_wilcox(), with FDR correction by the Benjamini–Hochberg method.

### Prediction of future T1D using the inflammatory proteome at birth

The diagnostic accuracy of several traditional and ML models—XGBoost, Random Forest, Logistic Regression, and SVM—for predicting future T1D based on the proteomic data is assessed in Python, using Jupyter Notebooks for reproducibility. By prioritizing proteins with high model-agnostic MI scores[82], the T1D prediction models focused on those proteins with the highest relative influence on prediction, capturing meaningful and potentially complex relationships in the proteins. Two methods for feature selection are taken in parallel, first using the top 40 proteins and then the set selected by the elbow method, applying a threshold for MI score. For the latter, MI scores are plotted on a histogram across different discretization levels and the point where the curve leveled off indicating the optimal feature selection, where additional information from other proteins does not add significantly to the reduction in uncertainty.

Training is carried out with 70% of the dataset (n = 275) and holding 30% for testing (n = 118), using as the selected protein NPX

values as predictors. For XGBoost, the baseline parameters are scale_pos_weight=4.0, random_state = 42. Stratified k-fold cross-validation is employed to maintain the same class distribution as seen in the original dataset (n_splits = 5, shuffle = True, random_state = 42). ROC curves are averaged across folds using interpolation on fixed set of FPRs. Performance metrics of accuracy, precision, recall, F1 score, and AUC-ROC are assessed. The best-performing model is identified and hyperparameter tuning performed using grid search with cross-validation. The following parameters are then tuned in the XGBoost model: scale_pos_weight = 10, max_depth = 3. Stratified k-fold cross-validation ensured balanced target distribution in eightfolds with a fixed random seed for reproducibility. The mean ROC curve is calculated along with a 95% confidence region.

Given the impact of DR4-DQ8 on T1D and its sufficient allele distribution in ABIS, the performance of ML and traditional models predicting future T1D individuals specifically with DR4-DQ8 alleles (n = 68) is assessed, in contrast to all controls (irrespective of HLA genotype, n = 268). Again, training is performed with 70% of the dataset and holding 30% for testing and a class weight multiplier of 4.2 (based on 68 T1D/286 controls). GridSearch is employed for hyperparameter tuning.

### Improved prediction with HLA and recursive feature elimination

To improve prediction, we implement another supervised ML pipeline using XGBoost, handling class imbalance via internal weighting as before by setting scale_pos_weight to mirror the ratio of controls to cases within the training data. However, here, feature selection is performed using recursive feature elimination (RFE) in scikit-learn, with XGboost as the base estimator, and HLA dosage is taken into account. RFE is optimal for optimizing feature selection in ensemble models, ranking features by contribution to model performance. The Olink case/control dataset (n = 432) is split into training (80%) and testing (20%) sets, stratified by outcome to preserve class proportions. RFE is applied to select the top 30 most predictive features, first excluding and then including HLA allele dosages (DR4DQ8 and DR3DQ2), extracting feature importances to assess their relative contributions. To evaluate model performance and generalizability while avoiding data leakage, we conduct fivefold stratified cross-validation using these same fixed selected features within each fold, preserving interpretability by consistently using this set of fixed features to see how they generalize throughout.

### Metabolites and T1D-linked proteins based on SHAP analysis

For 132 ABIS individuals for whom both metabolomic and proteomic data are available, non-linear protein-metabolite associations are determined using Shapley Additive exPlanations[24] (SHAP). For this analysis, HLA-DRA, IDS, and CTSC (higher in T1D), and ADA and TIMP3 (higher in controls), are tested. For each protein, feature selection is carried out by MI score for each of the proteins separately, by the elbow method to prioritize the top metabolites. Preliminary metabolite case/control differences are identified in prior analyses of the ABIS cohort[86,87]. As an expanded case set are not available at the time of this investigation to further evaluate these differences, differences are not shown here. Instead, our focus is on integration with proteomic data.

Proteomic associations with environmental factors are analyzed using a two-step approach. First, proteins are identified based on their statistical significance ($p < 0.05$) in previous analyses conducted across multiple age intervals (0–5, 6–10, 11–17, or 18–24 years), resulting in a total of 65 unique proteins that are still significant after FDR correction. After filtering to include only these proteins, Wilcoxon rank-sum tests are performed to evaluate associations between NPX levels for each protein and selected items on the birth questionnaires (e.g., stomach flu, smoking, severe life events during pregnancy, mode of delivery, and biological sex). For each protein, data are subsetted to include non-missing values and ensure valid group comparisons.

Medians of NPX levels for the two groups are calculated, and the Wilcoxon rank-sum test is used to determine whether distributions differed significantly. Multiple testing correction using the Benjamini–Hochberg (BH) method is employed to adjust for the false discovery rate (FDR). Results are compiled in a summary table, including the protein name, group medians, test statistic, $p$-value, and FDR-adjusted $p$-value. Proteins with FDR-adjusted $p$-values < 0.05 are considered significant.

### Ethical statement

All uses of human material have been approved. Ethical approval for ABIS is obtained by Research Ethics Committees of Faculty of Health Science at Linköping Univ., Ref. 1997/96287 and 2003/03-092 and the Medical Faculty of Lund University (Dnr 99227, Dnr 99321) prolongation of ABIS 03/092 and follow-up of adults 2019-05227) and connection to national registers (Dnr 03-513, 2018/380-32). All recruited volunteers in ABIS provided written informed consent, with participating parent(s) providing informed consent and later, the ABIS individuals themselves. Multinational collaborations with the Univ. of Florida are approved by the University of Florida's Institutional Review Board (IRB) as studies IRB201800903 and IRB202301239.

### Reporting summary

Further information on research design is available in the Nature Portfolio Reporting Summary linked to this article.

## Data availability

All processed proteomic, metabolomic, lipidomic, and associated metadata generated in this study are provided in the Source Data file. The raw metabolomic and lipidomic mass spectrometry data have been deposited in the Mass Spectrometry Interactive Virtual Environment (MassIVE) [https://massive.ucsd.edu/] under dataset identifier MSV000099985. These datasets constitute the minimum required information to reproduce and verify the analyses presented in this manuscript, while maintaining compliance with the ethical requirements governing human subjects research. Source Data are provided with this paper Source data are provided with this paper.

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

## Acknowledgements

We thank the ABIS participants and their families for participating in this research. We thank Ingela Johansson, Andrea Lebena, and Austeja Rutkauskaite for their help with sample distribution and for providing authors with register data. This work is supported by the Inflammation in human early life: targeting impacts on life-course health (INITIALISE) consortium funded by the Horizon Europe Program of the European Union under grant agreement 101094099 to M.O. and J.L. Further funding is granted to J.L. by Barndiabetesfonden (the Swedish Child Diabetes Foundation); Swedish Research Council (grant/award numbers K2005-72X-11242-11A, K2008-69X-20826-01-4, K2008-69X-20826-01-4); Medical Research Council of Southeast Sweden (FORSS); JDRF Wallenberg Foundation (grant/award no. K 98-99D-12813-01A); and ALF and LFoU grants from Region Östergötland and Linköping University, Sweden and Joanna Cocozza Foundation. This work used super-computing resources at the University of Florida Research Computing, provided by HiPerGator from NVIDIA.

## Author contributions

J.L. founded the ABIS study and designed the proteomic experiment. A.P.A. conceived the current study, with supervision from J.L., E.W.T., and R.D. and input from all authors. J.L. confirmed clinical diagnoses using the Swedish National Registry. ABIS coordinators, under the supervision of J.L., collected the questionnaire data. A.P.A. analyzed the data and prepared the results, with input from all authors and collaboration with R.D. on the machine learning models. A.P.A. wrote the first draft of the manuscript, incorporating feedback from all authors. T.H. and M.O. performed the metabolomics and lipidomics and conducted all relevant data processing. J.L. oversaw the research process, and J.L., A.P.A., and E.W.T. managed the ethical review process, with J.L. responsible for the foundational ABIS human subjects approvals. J.L. and M.O. secured funding and provided financial oversight. All authors had access to and verified the ABIS data. J.L., A.P.A., and E.W.T. had final responsibility for the decision to submit the manuscript for publication.

## Competing interests

The authors declare no competing interests.
