## [Transparent Peer Review file · Nature Communications]

The inflammatory path toward type 1 diabetes begins during pregnancy

Corresponding Author: Professor Eric Triplett

Version 0:

Reviewer comments:

Reviewer #1

(Remarks to the Author)

In this manuscript Ahrens et al performed Olink-based proteomics of cord serum samples to identify possible biomarkers that can predict the development of type 1 diabetes. They found inflammatory signatures that can predict the onset of the disease with good accuracy. The methodology is rigorous and described with sufficient details. I believe that this is an important finding and of interest for the scientific community. I only have a few very minor comments:

1. Protein "expression" is an incorrect term as expression refers to the transcription process. I would change to protein "abundance".
2. Line 267 is missing to close the parentheses.
3. Lines 285-286 – I think it should be a little clearer that the pathway is activated but not necessarily that there is an infection or that pathway is only activated by Yersinia.
4. The fonts used in figure 3 are too small and hard to read.

Reviewer #2

(Remarks to the Author)

I am really grateful to review this manuscript. In my opinion, this manuscript can be published once some revision is done successfully. I made two suggestions and I would like to ask your kind understanding.

Overview

The application of statistical approaches in type-1 diabetes (T1D) centers on logistic regression with small-scale data in clinical, demographic and/or invasive genetic contents. Little literature is available on the application of machine learning in T1D with large-scale data in non-invasive genetic contents including proteomic information. For this reason, this study attempted to evaluate the usefulness of machine learning as a predictive and explainable statistical approach regarding T1D with large-scale data in non-invasive proteomic contents. This study used proteomic data from 16683 children (78% of a general population), applied four machine learning models and achieved the area under the curve of 83% with boosting and the random forest.

This study presented boosting Shapley Additive Explanations (SHAP) variable importance outcomes as well, centering on 40 proteins such as HLA class II histocompatibility antigen, DR alpha chain (HLA-DRA), iduronate 2-sulfatase (IDS), secretoglobulin family 3A member 2 (SGB3A2), cathepsin C (CTSC), adenosine deaminase (ADA) and tissue inhibitor of metalloproteinases 3 (TIMP3). In addition, this study reported that boosting and the random forest registered the areas under the curves of 49%-52% based on clinical, demographic and environmental information. I would argue that this is a good start.

Suggestion 1: Boosting vs. Random Forest

However, firstly, it can be noted that boosting and the random forest often register similar performance outcomes but bring different results in permutation importance and SHAP plots. Boosting focuses on the final decision of the strongest tree with minimal bias whereas the random forest focuses on the majority vote of all trees with minimal variance. Both models have

their own strengths and weaknesses, hence which model performs and explains better depends on various conditions. In this vein, I would like to ask the authors to (1) compare their strengths and weaknesses and (2) compare their permutation importance and SHAP plots in a comprehensive manner.

Suggestion 2: Comprehensive Approaches to Integrate Various Types of Information

Secondly, it can be noted that this study requests due attention to machine learning with multi-modal data but hasn't outlined comprehensive approaches to integrate various types of information, i.e., clinical, demographic, environmental, proteomic and other genetic. The area under the curve of boosting and the random forest in this study, 83%, is above the minimal requirement of diagnostic criteria but still below its excellent standard. Are there some ways to improve this performance outcome by developing comprehensive approaches to integrate clinical, demographic, environmental, proteomic and other genetic information such as microRNA? If so, are there some ways to optimize across different forms of data expression so as to obtain more reliable performance and explanation results? I would like to ask the authors to address this issue in greater detail in the section of Discussion.

Reviewer #3

(Remarks to the Author)

The authors provide data on protein expression in cord blood samples obtained from a Swedish birth cohort of individuals who do or do not develop T1D. The data are likely to be useful to the field, but their analysis and interpretation here could be better described.

Major comments:

In general, there is too much verbiage regarding interpretation of pathway/GSEA analysis data. Pathway analysis can be very helpful for hypothesis generation but is most useful when backed up by functional or other information showing the validity of the mechanism at play. For example, much is made of a pathway associated with *Yersinia*; yersiniosis is relatively uncommon in Sweden and seems unlikely to be the direct cause of any inflammation observed here, especially in cord blood samples. The 7 proteins detected in the *Yersinia* pathway have a variety of roles in the immune system and can appear in multiple pathways. This is just one example of many pathways described in the manuscript. Without considerable further laboratory work to back up the pathway analysis, much of the speculation around all of the pathways discussed throughout the manuscript (results and discussion) should be toned back.

Conversely, the authors do not devote enough time to why a variety of methodological decisions were made in their project. For example, why did they use random selection – rather than selecting to ensure the cohorts were similar on important variables - when identifying the cohort for Olink measurement? Why were not all T1D cases included in the Olink work?

More clearly describing who is included in each analysis is key to this work. While there are a couple of analyses that included the full ABIS cohort, the majority of this paper uses smaller numbers of future T1D cases compared to controls. A visualization of cases and controls used across each analysis/figure would be helpful. All figures should include the n for cases and controls, and when subsetting is done (e.g. by age), the n for the subsets of interest should be provided.

In many cases, it is not clear how many variables were tested for a given analysis, nor what the cutoff for significance is and whether it differs by figure. For example, supplemental table 1 lists 12 features that have an adjusted $p < 0.05$. However, the number of variables tested from the birth questionnaires was not mentioned, the number of cases vs controls tested for each variable is not presented, and the cutoff for which of these variables are considered important to the authors is not stated. The text states that significant factors included “maternal illness in pregnancy” – but it is not clear which of these 12 features the authors refer to as related to maternal illness. The authors mention a significant difference by “medications in pregnancy” – is that referencing the use of psychotropics, or anesthesia during pregnancy? Odds ratios and directions are not provided in the table for these variables, making it challenging to assess.

The abstract mentions “toxic chemical exposure.” There is not direct evidence of this that I can find in the manuscript. Similarly, no detail on “stressful life events” or “environmental exposures” or “environmental factors.” Much more specificity is needed throughout the manuscript in order to contextualize the results. The discussion in particular should be largely re-written. Elements of the introduction are not sufficiently precise – for example: a) there is no broad international requirement that children be 2 years old for measurement of antibodies; b) this manuscript does not yield clear evidence that genetic or autoantibody screening could be avoided in our current understanding of T1D risk; c) the majority of the environmental factors described in the introduction have been variably associated and unassociated with disease depending on the cohort, but this nuance is not mentioned.

Supp Table 1 – why are some of the features missing odds ratios and directionality of the effect? Please provide the n for cases and controls separately rather than just the total.

Supp Table 3 – provides information on when the subjects selected for Olink differed from those not selected for Olink study, separated by cases and controls. An important additional piece of information is how the selected cases may or may not differ from the selected controls.

The methods do not describe where the metabolomics data were obtained from, nor how they were generated.

Minor comments:

Supp Table 8 has a formatting issue.

The methods section says that variables missing 15% or more of subjects were excluded, but the main text results says 50%.

Version 1:

Reviewer comments:

Reviewer #1

(Remarks to the Author)

The authors satisfactorily addressed all my concerns.

Reviewer #2

(Remarks to the Author)

I am really grateful to re-review this manuscript. In my opinion, this manuscript can be published in current form.

Reviewer #3

(Remarks to the Author)

The authors have made many welcome changes to the manuscript, but some questions remain.

My previous request for numbers of cases and controls to be added to all figures and analyses was only partially considered. Figure 3 is the most worrisome; the legend still seems to lump cases and controls together by age group, and the youngest age group is only 23 subjects. I respectfully ask again that this be added to every figure, including the schematic figure 1, so that readers can contextualize the findings presented. (Separately, the color scale on figure 3c and 3d make it impossible to see the numbers and asterisks, particularly for the proteins that are most significant.)

I continue to have concerns regarding the specificity of language in some aspects of the writing, including the introduction and discussion. An example concern is the section starting at line 81 ("Prenatal and environmental risk factors associated with future T1D"). Supplemental table 1 is the place this data are presented. A reader may assume that "prenatal and environmental risk factors" would mean chemical or pollutant exposures, parental smoking, viruses, or a broad set of possibilities. Instead, the only environmental factor listed in the main text as significant is "whether time has been spent with someone with insulin-dependent diabetes", which in a birth cohort study is likely correlated to the family history data also shown to be significant. I don't have a problem with the fact that few to no dietary, smoking, or other "traditional" environmental factors were identified in the chi-squared analysis. However, I would ask again for language that more precisely describes the manuscript's findings to be employed throughout the manuscript. (Separately, the methods and supplemental table 1 legend do not indicate that multiple testing correction was carried out for the chi squared analysis in that section; if not performed this should be added.)

The introduction and discussion use many words to say why genetic risk and autoantibody testing are insufficient. This may well be the case. However: the authors here present a single (very interesting!) study of cord blood proteomics in people at risk for T1D. This work is interesting on its own – the introduction and discussion do not need to describe the costs or controversies of genetic or autoantibody testing (which is not mentioned for proteomics and metabolomics) in order for their findings to be interesting. The manuscript provides no evidence that this cord blood test would be sufficient to identify when a person would get diabetes – which is information that change in autoantibodies and eventually glucose monitoring would provide. To me, all of that is fine – there is no reason that this single cord blood dataset needs to improve upon decades of genetic and autoantibody data in order to be interesting to this audience. However, the corresponding language should be toned back.

Similarly, the teplizumab component of the discussion is irrelevant to the work presented here. I agree the use of this therapy is rightfully controversial, as it is both expensive and imperfect. However, this study was not designed to shed new light on when/whether/how teplizumab should be used, meaning lines 276 through 293 could all be cut from the discussion. Figure 2C: It is impossible to see in the figure whether some of the odds ratios cross 1. Can the actual range numbers be provided as part of the image? Also, supplemental table 1 suggests that having a mother born in Sweden was more common in those children developing T1D vs not (~98% of cases vs ~93% of controls), similar in direction to having a mother with T1D (~7% of cases vs 0.7% of controls) . However, the odds ratio plot in 2c seems to suggest that having a mother born in Sweden is protective against T1D in this study. Can the authors comment on this? Why not include all of the factors shown to be important (2a and 2b) as odds ratios (2c)?

I continue to feel that much of the discussion could be reduced in word count. Many sentences explain what the proteins identified as important may do when properly located on, inside, or near immune cells or beta cells. However, the relationship between protein concentration in cord blood and these normal cellular functions is not demonstrated, and a role for some of them as secreted or cleaved proteins (e.g. HLA-DRA) is unclear or unstated. This is ok – but substantially reducing the amount of words from lines 311-373 dedicated to these processes – particularly beta cell processes which may

or may not be strongly reflected in cord blood given their small number in both mother and fetus - is warranted.

The non-parametric correlations between proteins and metabolites in extended data figure 9 appear driven by only a few outlying subjects. Do other methods of correlation (standard Spearman's correlations which are also non-parametric, perhaps) show that these are significant relationships? Separately, the compounds on the X-axis of the second row of figure part a are cut off and not included in the legend.

While mutual information scores and SHAP values for model outputs are provided for many of the metabolomic measures, actual raw differences between cases and controls for the variables themselves are not presented. This should be added to contextualize findings.

The authors frequently report findings that are not significant after multiple testing correction. Given the very large number of tests performed in this manuscript, it seems that only features passing multiple test correction should be discussed throughout the manuscript. For example, extended figure 10 – and discussion lines 400-404 – all suggest that extended figure 10 identified a link between toxic exposures, protein levels, and T1D – but none of these data met false discovery correction. This should all be removed, as well as the nearby statement about dietary risk, which is in fact not backed up by the environmental questionnaire data in this study.

Line 411: please strike the word “more” before evidence of generalizability. This needs to be tested in at least one large independent cohort, which is a reasonable caveat to state simply and clearly.

Version 2:

Reviewer comments:

Reviewer #3

(Remarks to the Author)

1) My previous request for changes regarding the specificity of language was in many places not addressed. Examples include:

a) The same section mentioned “Prenatal risk factors associated with T1D” still contains language that is unspecific – the “prenatal diet” aspect refers specifically to low fried potato consumption. The full list of environmental factors (n=44) studied is not presented in Supplemental Table 1 for the reader to understand what other factors were included – how did fried potato consumption end up in this list of only 44 variables? Returning to my review of the original version of the manuscript – odds ratios are not provided for all variables but it appears that the data suggest increased fried potato consumption is protective against T1D?

b) “Prenatal and perinatal factors” are also non-specifically discussed in the abstract.

c) The manuscript mentions “environmental factors” throughout – including in the first and second paragraphs of the introduction. The PFOS-L data are not presented sufficiently to justify the level of discussion here, and their raw values were not presented in any figure though this was requested so that the reader can see how strong the association may be between PFOS and disease development. PFOS-L levels are also not shown to be associated with proteins in extended figure 9.

2) Similarly, my previous request regarding the comparison of the predictive abilities of cord blood proteomics to genetics, antibody testing, or other “invasive procedures” was also only partially addressed – this concept still remains as the final sentence of the introduction, as well as other locations including throughout the discussion. The authors have not identified a biomarker nor anything close to as predictive as genetics or autoantibodies.

3) Changes regarding false-discovery were welcome. However I note that the authors still state these findings in the “Prenatal/perinatal factors and T1D-associated proteins” section - they just then state the data are not shown. This sentence should be removed.

Version 3:

Reviewer comments:

Reviewer #3

(Remarks to the Author)

The data regarding PFOS-L were provided in supplemental materials. Optimally since the authors state that there is a non-linear relationship, the authors would provide at least one new plot indicating the relationship between levels of any of their proteins of interest and levels of PFOS-L. Optimally the authors would also provide an explanation why they focus on those 43 variables from the ABIS questionnaire.

RESPONSE TO REVIEWERS' COMMENTS

Reviewer #1:

I only have a few very minor comments:

1. Protein “expression” is an incorrect term as expression refers to the transcription process. I would change to protein “abundance”.

An excellent point. Protein "expression" has been changed to protein "concentration" or "abundance" throughout the text. In Olink proteomics, the values correspond to Normalized Protein eXpression (NPX). However, this metric really indicates a relative abundance or concentration within the sample.

2. Line 267 is missing to close the parentheses.

A few typographical errors, including this closed parenthesis, were noted in our revision and have been addressed. Thank you!

3. Lines 285-286 – I think it should be a little clearer that the pathway is activated but not necessarily that there is an infection or that pathway is only activated by Yersinia.

Agreed. We now mention that network representation within this pathway does not necessarily translate to an active or past Yersinia infection. Constituents of this pathway are activated not only by Yersinia but could have other origins.

4. The fonts used in figure 3 are too small and hard to read.

We agree – the fonts of the enrichment plots were very small as presented originally. Thank you for noting this. The fonts are now much larger in the revised version. These plots have been moved to an Extended Data Figure (Extended Data Fig. 3) for enhanced visibility.

Reviewer #2:

Suggestion 1: Boosting vs. Random Forest

However, firstly, it can be noted that boosting and the random forest often register similar performance outcomes but bring different results in permutation importance and SHAP plots. Boosting focuses on the final decision of the strongest tree with minimal bias whereas the random forest focuses on the majority vote of all trees with minimal variance. Both models have their own strengths and weaknesses, hence which model performs and explains better depends on various conditions. In this vein, I would like to ask the authors to (1) compare their strengths and weaknesses and (2) compare their permutation importance and SHAP plots in a comprehensive manner.

Thank you – yes, the features driving random forest and boosting methods may differ. This is an excellent point. For this reason, we first performed mutual information scoring to extract relevant features (as a common ground) and then applied the comparative ML and traditional models for prediction using those features alone in the original analysis. We hope that this point is clearer in the current draft. We have largely rewritten this section of the results, reflecting improved models (addressed in other feedback). Given the extensive additional machine learning models that have been added to this revision of the manuscript (showing much improved performance from the previous draft), we have moved the other material from the preliminary models to Extended Data.

Suggestion 2: Comprehensive Approaches to Integrate Various Types of Information

Secondly, it can be noted that this study requests due attention to machine learning with multi-modal data but hasn't outlined comprehensive approaches to integrate various types of information, i.e., clinical,

demographic, environmental, proteomic and other genetic. The area under the curve of boosting and the random forest in this study, 83%, is above the minimal requirement of diagnostic criteria but still below its excellent standard. Are there some ways to improve this performance outcome by developing comprehensive approaches to integrate clinical, demographic, environmental, proteomic and other genetic information such as microRNA? If so, are there some ways to optimize across different forms of data expression so as to obtain more reliable performance and explanation results? I would like to ask the authors to address this issue in greater detail in the section of Discussion.

This was a very important suggestion. Thank you for this feedback, which prompted us to build more comprehensive prediction models for the revision. In doing so, an AUC of 0.89 was achieved – much improved from the original draft. The model accounts for HLA genetics, and we present the nuances of this in the discussion. We have not incorporated other data (such as demographics and environment), as the goal of our investigation is to present the most simple, non-invasive way to predict future disease that could be easily and inexpensively added to clinical practice at birth. Although these features may improve T1D prediction within this Swedish cohort, it would also likely reduce the generalizability to other populations, where these characteristics differ drastically due to culture or geography.

Reviewer #3:

Major comments:

In general, there is too much verbiage regarding interpretation of pathway/GSEA analysis data. Pathway analysis can be very helpful for hypothesis generation but is most useful when backed up by functional or other information showing the validity of the mechanism at play. For example, much is made of a pathway associated with *Yersinia yersiniosis* is relatively uncommon in Sweden and seems unlikely to be the direct cause of any inflammation observed here, especially in cord blood samples. The 7 proteins detected in the *Yersinia* pathway have a variety of roles in the immune system and can appear in multiple pathways. This is just one example of many pathways described in the manuscript. Without considerable further laboratory work to back up the pathway analysis, much of the speculation around all of the pathways discussed throughout the manuscript (results and discussion) should be toned back.

Agreed. The speculation on these pathways has been toned backed and the language is now more concise. Moreover, we have provided more extensive literature and discussion surrounding what is known about the most significant proteins and how this may play a role in stress of the beta cells, ultimately leading to T1D. We hope that you find the discussion is much improved.

Conversely, the authors do not devote enough time to why a variety of methodological decisions were made in their project. For example, why did they use random selection – rather than selecting to ensure the cohorts were similar on important variables - when identifying the cohort for Olink measurement? Why were not all T1D cases included in the Olink work?

The controls and T1D cases used in this study were indeed largely representative of those in the full ABIS cohort. We appreciate this point, as it prompted us to present a side-by-side comparison of significance across important child characteristics (now Extended Data Fig. 2), comparing those for whom we have proteomic data vs. those for whom we do not. The only notable difference was a modest overrepresentation of females among controls included in the Olink subset (52.5%) versus those controls who were not (47.9%). As many T1D cases as was practical to process for proteomics were included in this investigation.

More clearly describing who is included in each analysis is key to this work. While there are a couple of analyses that included the full ABIS cohort, the majority of this paper uses smaller numbers of future T1D cases compared to controls. A visualization of cases and controls used across each analysis/figure would

be helpful. All figures should include the n for cases and controls, and when subsetting is done (e.g. by age), the n for the subsets of interest should be provided.

We appreciate this comment. Transparency of the sample size is very important to us, as well. To address your feedback, we have gone through every major analysis presented in this paper and the corresponding figures to ensure that the sample size is presented.

In many cases, it is not clear how many variables were tested for a given analysis, nor what the cutoff for significance is and whether it differs by figure. For example, supplemental table 1 lists 12 features that have an adjusted $p < 0.05$. However, the number of variables tested from the birth questionnaires was not mentioned, the number of cases vs controls tested for each variable is not presented, and the cutoff for which of these variables are considered important to the authors is not stated. The text states that significant factors included “maternal illness in pregnancy” – but it is not clear which of these 12 features the authors refer to as related to maternal illness.

Thank you for bringing up this point. For transparency, we have completely reworked the original Supplementary Table. We have also included the overview of the items tested in the results section. In the new table, the “n” is presented for both cases and controls, along with the p value. A total of 44 variables were tested from the birth questionnaire, and these are presented.

The authors mention a significant difference by “medications in pregnancy” – is that referencing the use of psychotropics, or anesthesia during pregnancy? Odds ratios and directions are not provided in the table for these variables, making it challenging to assess.

There were two medications during pregnancy that were significant at this comparative level: antibiotics and psychotropics, so this is now indicated more clearly. To achieve greater transparency and clarity, Supplementary Table 1 has been completely replaced to show the original chi-square statistics.

The abstract mentions “toxic chemical exposure.” There is not direct evidence of this that I can find in the manuscript. Similarly, no detail on “stressful life events” or “environmental exposures” or “environmental factors.” Much more specificity is needed throughout the manuscript in order to contextualize the results.

The association of the proteins with these factors is described in the “Metabolites, exogenous toxic compounds, and T1D-linked proteins” and “Prenatal/perinatal factors and T1D-associated proteins” paragraphs in the Results section. This has been revised in the abstract to maintain brevity, with more specific mention of a few notable cord blood compounds associated with the proteins of interest. Extended Data Fig. 9 addresses associations with cord blood metabolites and exogenous compounds; Figure 6 and Extended Data Fig. 10 address associations with prenatal and perinatal factors.

The discussion in particular should be largely re-written. Elements of the introduction are not sufficiently precise – for example: a) there is no broad international requirement that children be 2 years old for measurement of antibodies; b) this manuscript does not yield clear evidence that genetic or autoantibody screening could be avoided in our current understanding of T1D risk; c) the majority of the environmental factors described in the introduction have been variably associated and unassociated with disease depending on the cohort, but this nuance is not mentioned.

Thank you for this feedback. We have expanded (and largely rewritten) the discussion to satisfy the reviewers’ points on these and other important caveats. Going beyond the initial constraint of the word limitations has allowed us to expand this discussion so it can be more informative. Additional citations have also been added.

Supp Table 1 – why are some of the features missing odds ratios and directionality of the effect? Please provide the n for cases and controls separately rather than just the total.

Supplementary Table 1 has been completely redone, showing now the “n” for cases and controls separately, as well as the p values from the chi-square tests of association for each tested variable of interest in the birth questionnaire.

Supp Table 3 – provides information on when the subjects selected for Olink differed from those not selected for Olink study, separated by cases and controls. An important additional piece of information is how the selected cases may or may not differ from the selected controls.

The controls were selected randomly from all controls with Olink data. No matching was done in the selection of controls. However, in paragraph two of the second “cord blood proteomic analysis”, we outline where the T1D subjects for Olink aligned with those of the full ABIS cohort. The slight difference in sex and gestational age is addressed in the text. The selected samples were broadly representative of the full cohort.

The methods do not describe where the metabolomics data were obtained from, nor how they were generated.

Thank you for noting this. We now include a section on metabolomics data, where we present the major details for generating these data, as well as a citation of a previous paper from our team with more specifics.

Minor comments:

Supp Table 8 has a formatting issue.

Thank you for reviewing these tables and finding this error. We have corrected Supplementary Table 8.

The methods section says that variables missing 15% or more of subjects were excluded, but the main text results says 50%.

This has been rewritten and Supplementary Table 1 reanalyzed for a better representation.

Sincerely,

Eric W. Triplett
Professor and Chair

Responses to reviewers for 2nd resubmission of manuscript number NCOMMS-25-04817B:

We consider it particularly important that you ensure that the number of cases and controls is clearly reported in all figures and analyses. In addition, carefully revise the text to ensure the specificity of the language throughout, as detailed by ref#3. Meanwhile, further edit the discussion section, as indicated by ref#3, and emphasize throughout the manuscript only features that pass multiple test corrections.

When evaluating your revised manuscript, we will not consider any similar papers published independently in the meantime to compromise the novelty of your study. See here for more information.

Reviewer #3 points:

- Regarding sample size
 - We have reviewed every figure and subpanel to ensure that the “n” is indicated. Many figures already had the “n” indicated, including Fig. 1, 2e, 3, 4, 5, 6a-f, and all Extended Data Fig. (except for Fig. 3, where the “n” is not applicable). We have clarified the “n” in Fig. 2a-d, where the case and control sample sizes are additionally provided in Supplementary Table 1. For Fig. 6g-h, the total “n” is indicated, and this reflects the size of the full Olink subset in our study.
- *“Separately, the color scale on figure 3c and 3d make it impossible to see the numbers and asterisks, particularly for the proteins that are most significant.”*
 - Thank you for the note on readability. Visibility here is very important. Fig. 3c and 3d have been generated with a new color scheme to improve readability, with white text against the darker cells (in either direction), and all labels were placed in bold.
- Regarding introduction and discussion (language): “Check specificity of language in some aspects of the writing, including the introduction and discussion. An example concern is the section starting at line 81 (“Prenatal and environmental risk factors associated with future T1D”). **Supplemental table 1** is the place this data are presented. A reader may assume that “prenatal and environmental risk factors” would mean chemical or pollutant exposures, parental smoking, viruses, or a broad set of possibilities. Instead, the only environmental factor listed in the main text as significant is “whether time has been spent with someone with insulin-dependent diabetes”, which in a birth cohort study is likely correlated to the family history data also shown to be significant. I don’t have a problem with the fact that few to no dietary, smoking, or other “traditional” environmental factors were identified in the chi-squared analysis. However, I would ask again for language that more precisely describes the manuscript’s findings to be

employed throughout the manuscript. (Separately, the methods and supplemental table 1 legend do not indicate that multiple testing correction was carried out for the chi squared analysis in that section; if not performed this should be added.”

- Our mention of “environmental factors” has been toned back or clarified, including in this paragraph of the Results. Thank you for pointing this out. The associations with environmental factors in this investigation and this approach were indeed rather limited (not very strong).
- We reworked Supp Table 1 for more clarity. FDR adjusted p values were provided in earlier draft, but now the n’s (for case and control) are clearly indicated and dash marks in the OR (95% CI) columns where the degrees of freedom exceeded 1, i.e., no odds ratio calculated.
- Regarding introduction and discussion (genetic risk and autoantibodies): “introduction and discussion use many words to say why genetic risk and autoantibody testing are insufficient. This may well be the case. However: the authors here present a single (very interesting!) study of cord blood proteomics in people at risk for T1D. This work is interesting on its own – the introduction and discussion do not need to describe the costs or controversies of genetic or autoantibody testing (which is not mentioned for proteomics and metabolomics) in order for their findings to be interesting. The manuscript provides no evidence that this cord blood test would be sufficient to identify when a person would get diabetes – which is information that change in autoantibodies and eventually glucose monitoring would provide. To me, all of that is fine – there is no reason that this single cord blood dataset needs to improve upon decades of genetic and autoantibody data in order to be interesting to this audience. However, the corresponding language should be toned back.”
 - We agree with the Reviewer and have significantly toned back this language. The edits are marked with track changes enabled in the revised draft.
- Regarding Teplizumab: “irrelevant to the work presented here. I agree the use of this therapy is rightfully controversial, as it is both expensive and imperfect. However, this study was not designed to shed new light on when/whether/how teplizumab should be used, meaning lines 276 through 293 could all be cut from the discussion.”
 - These sections have been cut accordingly, and we feel that the discussion reads more focused with this change.
- Regarding Fig. 2c and Supplemental Table 1: “It is impossible to see in the figure whether some of the odds ratios cross 1. Can the actual range numbers be provided as part of the image? Also, supplemental table 1 suggests that having a mother born in

Sweden was more common in those children developing T1D vs not (~98% of cases vs ~93% of controls), similar in direction to having a mother with T1D (~7% of cases vs 0.7% of controls). However, the odds ratio plot in 2c seems to suggest that having a mother born in Sweden is protective against T1D in this study. Can the authors comment on this? Why not include all of the factors shown to be important (2a and 2b) as odds ratios (2c)?

- We have revised Fig. 2c accordingly. ORs and their 95% CI are now indicated in the figure, as well. (Only the factors from Supplementary Table 1 with a single degree of freedom are included in the forest plot.) We appreciate Reviewer 3's attention to detail here so we could correct this.
- Regarding discussion word count: "much of the discussion could be reduced in word count. Many sentences explain what the proteins identified as important may do when properly located on, inside, or near immune cells or beta cells. However, the relationship between protein concentration in cord blood and these normal cellular functions is not demonstrated, and a role for some of them as secreted or cleaved proteins (e.g. HLA-DRA) is unclear or unstated. This is ok – but **substantially reducing the amount of words** from lines 311-373 dedicated to these processes – particularly beta cell processes which may or may not be strongly reflected in cord blood given their small number in both mother and fetus - is warranted."
 - We have significantly reduced the discussion to improve focus while still providing biological rationale for the findings.
- Regarding Extended Data Fig. 9: "non-parametric correlations between proteins and metabolites in extended data figure 9 appear driven by only a few outlying subjects. Do other methods of correlation (standard Spearman's correlations which are also non-parametric, perhaps) show that these are significant relationships? Separately, **the compounds on the X-axis of the second row of figure part a** are cut off and not included in the legend."
 - We appreciate the reviewer's point. Although Spearman is nonparametric, it is not immune to outliers entirely as they can still affect the rank order. To address this concern, we have removed the most extreme 1-2% of samples, retested with the Spearman, and compared results to the original test. We tested this by removing the top 2.5% and bottom 2.5% (thus, the 5% of extremes), rather than a 1-2.5% given the smaller sample size of the metabolomic dataset. After removing the top and bottom 1% of values for each variable similar results were found, confirming that our findings at least for TIMP3 were not driven by

extreme values (although some protein x metabolite pairings did drop out once the extremes were removed). For TIMP3, a few associations fell away, so we swapped the panel in the figure. For NTF3, the association went away, so we removed the panel. For PDLIM7, most associations fell away, so we removed the panel. We added text to the results to contextualize this approach and the corresponding results for transparency to the reader.

- Regarding raw differences in metabolites: “mutual information scores and SHAP values for model outputs are provided for many of the metabolomic measures, actual raw differences between cases and controls for the variables themselves are not presented. This should be added to contextualize findings.”
 - As the raw differences in the metabolites was not the purpose of the present study, but rather published in two earlier ABIS reports, we do not present those data here. If the dataset had been expanded and we could include those data here, we would have, but unfortunately a more extensive sample size is not available at this time. Instead, we are focused here on how these metabolites align with the proteins most relevant to prediction. This is an important point that Reviewer 3 raises, so now we need to refer readers to those papers to see the metabolomic case/control results.
- Regarding findings not significant after FDR: “Given the very large number of tests performed in this manuscript, it seems that only features passing multiple test correction should be discussed throughout the manuscript. For example, extended figure 10 – and discussion lines 400-404 – all suggest that extended figure 10 identified a link between toxic exposures, protein levels, and T1D – but none of these data met false discovery correction. **This should all be removed, as well as the nearby statement about dietary risk**, which is in fact not backed up by the environmental questionnaire data in this study.”
 - We have removed Extended Data Fig. 10 since it was only significant before FDR. We also removed this in text: “Associations of smoking by the mother during pregnancy, maternal stomach flu during pregnancy, and severe life event during pregnancy were only significant before FDR correction.”
- Regarding discussion sentence: “Line 411: please strike the word “more” before evidence of generalizability. This needs to be tested in at least one large independent cohort, which is a reasonable caveat to state simply and clearly.”
 - We have made this change in the present text. We agree, this is a reasonable consideration.

We thank all Reviewers for their careful feedback, which has significantly improved the content and messaging of this paper.

RESPONSES TO THE REVIEW

Ref#3 still raises concerns about the specificity of the language, especially regarding the link between environmental factors and T1D development and the ability to detect this relationship in cord blood.

The language has been revised throughout to address this point from Reviewer 3.

Reviewer #3 (Remarks to the Author)

1) My previous request for changes regarding the specificity of language was in many places not addressed. Examples include:

a) The same section mentioned “Prenatal risk factors associated with T1D” still contains language that is unspecific – the “prenatal diet” aspect refers specifically to low fried potato consumption. The full list of environmental factors (n=44) studied is not presented in Supplemental Table 1 for the reader to understand what other factors were included – how did fried potato consumption end up in this list of only 44 variables? Returning to my review of the original version of the manuscript – odds ratios are not provided for all variables but it appears that the data suggest increased fried potato consumption is protective against T1D?

Supplemental Table 1 has been revised for clarity and the source data for figures is now provided as a companion to the manuscript, including odds ratios.

b) “Prenatal and perinatal factors” are also non-specifically discussed in the abstract.

The abstract has been revised to specify delivery mode.

c) The manuscript mentions “environmental factors” throughout – including in the first and second paragraphs of the introduction. The PFOS-L data are not presented sufficiently to justify the level of discussion here, and their raw values were not presented in any figure though this was requested so that the reader can see how strong the association may be between PFOS and disease development. PFOS-L levels are also not shown to be associated with proteins in extended figure 9.

For point 1, the language has been simplified at every point requested. Supplemental Table 1 (the attached excel file) has been corrected and improved as requested.

2) Similarly, my previous request regarding the comparison of the predictive abilities of cord blood proteomics to genetics, antibody testing, or other “invasive procedures” was also only partially addressed – this concept still remains as the final sentence of the introduction, as well as other locations including throughout the discussion. The authors have not identified a biomarker nor anything close to as predictive as genetics or autoantibodies.

For point, 2, all sentences regarding replacing the biomarkers we have discovered in this cohort as a prevention tool over genetics and autoantibody assays have been removed from the Introduction and the Discussion.

3) Changes regarding false-discovery were welcome. However I note that the authors still state these findings in the “Prenatal/perinatal factors and T1D-associated proteins” section - they just then state the data are not shown. This sentence should be removed.

For point 3, the offending sentence was removed.

Please also include the reporting summary and provide a Source data file (by which we mean an Excel table containing the numerical data based on which the graphs were drawn.)

Thank you for the comment. Source Data has been compiled for every figure and is now provided so that the manuscript follows this requirement. A Reporting Summary file was also prepared.